# BAYESIAN EXPLORATION FOR LIFELONG REINFORCEMENT LEARNING

## ABSTRACT

A central question in reinforcement learning (RL) is how to leverage prior knowledge to accelerate learning in new tasks. We propose a Bayesian exploration method for lifelong reinforcement learning (BLRL) that aims to learn a Bayesian posterior that distills the common structure shared across different tasks. We further derive a sample complexity analysis of BLRL in the finite MDP setting. To scale our approach, we propose a variational Bayesian Lifelong Learning (VBLRL) algorithm that is based on Bayesian neural networks, can be combined with recent model-based RL methods, and exhibits backward transfer. Experimental results on three challenging domains show that our algorithms adapt to new tasks faster than state-of-the-art lifelong RL methods.

## 1 INTRODUCTION

Reinforcement-learning (RL) methods (Sutton & Barto, 1998; Kaelbling et al., 1996) have been successfully applied to solve challenging individual tasks such as learning robotic control (Duan et al., 2016) and playing expert-level Go (Silver et al., 2017). However, in the real world, a robot usually experiences a collection of distinct tasks that arrive sequentially throughout its operational lifetime; learning each new task from scratch is infeasible, but treating them all as a single task will fail. Therefore, recent research has focused on algorithms that enable agents to learn across multiple, sequentially posed tasks, leveraging past knowledge from previous tasks to accelerate the learning of new tasks. This problem setting is known as *lifelong reinforcement learning* (Brunskill & Li, 2014; Wilson et al., 2007b; Isele et al., 2016b). The key questions in lifelong RL research are: How can an algorithm exploit knowledge gained from past tasks to improve performance in new tasks (forward transfer), and how can data from new tasks help the agent to perform better on previously learned tasks (backward transfer)?

To answer these two questions, first consider a simple problem, which is to find different items in different houses. Here, a single task corresponds to finding items in a specific house. Although items may be stored in different locations in different houses, there still exists some shared information that connects all houses. For instance, a toothbrush is more likely to be found in a bathroom than a kitchen, and a room without a window is more likely to be a bathroom than a living room. Such information can significantly accelerate the search for items in newly encountered houses. We propose that extracting the common structure existing in previously encountered tasks can help the agent quickly learn the dynamics of the new tasks. Specifically, this paper considers lifelong RL problems that can be modeled as hidden-parameter MDPs or *HiP-MDPs* (Doshi-Velez & Konidaris, 2016; Killian et al., 2017), where variations among the true task dynamics can be described by a set of hidden parameters. We model two main categories of learning across multiple tasks: the world-model distribution, which describes the probability distribution over tasks, and the task-specific model, that defines the (stochastic) dynamics within a single task. To enable more accurate sequential knowledge transfer, we separate the learning process for these two quantities and maintain a hierarchical Bayesian posterior to approximate them. The world-model posterior is designed to manage the uncertainty in the world-model distribution, while the task-specific posterior handles the uncertainty from the data collected from only the current task.

We propose a Bayesian exploration method for lifelong RL (BLRL) that learns a Bayesian world-model posterior that distills the common structure of previous tasks, and then uses it as a prior to learn a task-specific model in each subsequent task. For the discrete case, we derive an explicit

performance bound that shows that the task-specific model requires fewer samples to become accurate as the world-model posterior approaches the true underlying world-model distribution. We further develop VBLRL, a more scalable version of BLRL that uses variational inference to approximate the world-model distribution and leverages Bayesian Neural Networks (BNNs) to build the hierarchical Bayesian posterior. Our experimental results on a set of challenging domains show that our algorithms achieve better forward and backward transfer performance than state-of-the-art lifelong RL algorithms within limited samples for each task.

## 2 BACKGROUND

RL is the problem of maximizing the long-term expected reward of an agent interacting with an environment (Sutton & Barto, 1998). We usually model the environment as a Markov Decision Process or *MDP* (Puterman, 1994), described by a five tuple: $\langle S, A, R, T, \gamma \rangle$, where $S$ is a finite set of states; $A$ is a finite set of actions; $R : S \times A \mapsto [0, 1]$ is a reward function, with a lower and upper bound 0 and 1; $T : S \times A \mapsto \Pr(S)$ is a transition function, with $T(s'|s, a)$ denoting the probability of arriving in state $s' \in S$ after executing action $a \in A$ in state $s$; and $\gamma \in [0, 1)$ is a discount factor, expressing the agent's preference for delayed over immediate rewards.

An MDP is a suitable model for the task facing a single agent. In the lifelong RL setting, the agent instead faces a series of tasks $\tau_1, ..., \tau_n$, each of which can be modeled as an MDP: $\langle S^{(i)}, A^{(i)}, R^{(i)}, T^{(i)}, \gamma^{(i)} \rangle$. A key question is how these task MDPs are related; we model the collection of tasks as a HiP-MDP (Doshi-Velez & Konidaris, 2016; Killian et al., 2017), where a family of tasks is generated by varying a latent task parameter $\omega$ drawn for each task according to the world-model distribution $P_\Omega$. Each setting of $\omega$ specifies a unique MDP, but the agent neither observes $\omega$ nor has access to the function that generates the task family. Formally, then, the dynamics $T(s'|s, a; \omega_i)$ and reward function $R(r|s, a; \omega_i)$ for task $i$ depend on $\omega_i \in \Omega$, which is fixed for the duration of the task. For lifelong RL problems, the performance of a specific algorithm is usually evaluated based on both forward transfer and backward transfer results (Lopez-Paz & Ranzato, 2017):

- *Forward transfer*: the influence that learning task $t$ has on the performance in future task $k \succ t$.

- *Backward transfer*: the influence that learning task $t$ has on the performance in earlier tasks $k \prec t$.

## 3 BAYESIAN EXPLORATION FOR LIFELONG REINFORCEMENT LEARNING

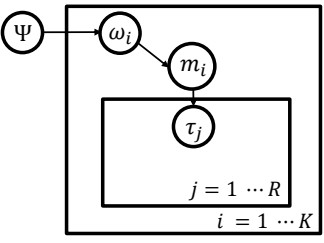

Figure 1: Plate representation for the BLRL approach. $\tau_j$ denotes trajectory $\{s, a, r, s'\}_j$. There are $K$ different tasks and the agent samples $R$ trajectories from each task.

The key component of our approach is a hierarchical Bayesian posterior over task MDPs controlled by the hidden parameter $\omega$. Intuitively, we maintain probability distributions that separately capture two categories of uncertainty within lifelong learning tasks: the world-model posterior captures the epistemic uncertainty of the world-model distribution over different tasks controlled by the hidden parameter. As the learner is exposed to more and more tasks, this posterior should converge to the world-model distribution $P_\Omega$. The task-specific posterior captures the epistemic uncertainty of the current task $m$. As the learner is exposed to more and more transitions within the task, this posterior should approach $m$, leaving only the aleatoric uncertainty of transitions within the task, which is independent of other tasks. By sampling from the world-model posterior, an agent can learn new tasks faster by exploiting knowledge common to previous tasks, thus exhibiting positive forward transfer.

We first consider the finite MDP setting. Concretely, we use a hierarchical Bayesian model to represent the distribution over MDPs. Figure 1 shows our generative model in plate notation. $\Psi$ is the parameter set that represents distribution $P_\Omega$. It functions as the world-model posterior that aims to capture the common structure across different tasks. The resulting MDP $m_i$ is created based on $\omega_i$, which is one hidden parameter sampled from $\Psi$. We can sample from our approximation of $\Psi$ to create and solve possible MDPs.

The proposed BLRL approach is formalized in Algorithm 3 in the appendix. Initially, before any MDPs are experienced, the world-model posterior $q_e(\cdot|s_t, a_t)$ is initialized to an uniformed prior. For each new task $m_i$, we first initialize the task-specific posterior $q_\theta^{m_i}(\cdot|s_t, a_t)$ with the parameter values from the current world-model posterior, and then, for each timestep, select actions using a Bayesian exploration algorithm based on sampling from this posterior (Thompson, 1933; Asmuth et al., 2009). A set of sampled MDPs drawn from $q_\theta^{m_i}$ is a concrete representation of the uncertainty within the current task. BLRL samples $K$ models from the task-specific posterior whenever the number of transitions from a state–action pair has reached threshold $B$. Analogously to RMAX (Brafman & Tennenholtz, 2003) and BOSS (Asmuth et al., 2009), we call a state–action pair **known** whenever it has been observed $N_{s_t, a_t} = B$ times. For each state–action pair, if it is **known**, we use the task-specific posterior to sample the model. If it is **unknown**, we instead sample from the world-model posterior. These models are combined into a merged MDP $m_i^\#$ and BLRL solves $m_i^\#$ to get a policy $\pi^*_{m_i^\#}$. This approach is adopted from BOSS (best of sampled set) to create optimism in the face of uncertainty, and thereby drive exploration. The new policy $\pi^*_{m_i^\#}$ will be used to interact with the environment until a new state–action pair reaches the sampling threshold. The collected transitions from this task will be used to update the task-specific posterior immediately, while the world-model posterior will be updated using transitions from all the previous tasks at a slower pace. For simple finite MDP problems in practice, we use the Dirichlet distribution (the conjugate for the multinomial) to represent the Bayesian posterior. Thus, the updating process for the posterior is straightforward to compute. Intuitively, BLRL is able to rapidly adapt to new tasks as long as the prior of the task-specific model (that is, the world-model posterior) is close to the true underlying model and captures the uncertainty of the common structure of a set of tasks.

## 3.1 SAMPLE COMPLEXITY ANALYSIS

We now provide a simple theoretical analysis of BLRL. First, we use the setting and results of Zhang (2006) to describe the properties of the Bayesian prior and how it relates to the sample complexity for the concentration of the Bayesian posterior.

**Lemma 1.** Let $\pi(\omega)$ denote the prior distribution on the parameter space $\Gamma$. We consider a set of transition-probability densities $p(\cdot|\omega)$ indexed by $\omega$, and the true underlying density $q$. Define the **prior-mass radius** of the transition-probability densities as:

$$d_\pi = \inf\{d : d \geq -\ln \pi(\{p \in \Gamma : D_{KL}(q||p) \leq d\})\}. \tag{1}$$

Intuitively, this quantity measures the distance between the Bayesian prior we use to initialize the posterior and the true underlying distribution. Then, $\forall \rho \in (0, 1)$ and $\eta \geq 1$, let

$$\varepsilon_n = (1 + \frac{1}{n})\eta d_\pi + (\eta - \rho)\varepsilon_{upper,n}((\eta - 1)/(\eta - \rho)), \tag{2}$$

where $\varepsilon_{upper,n}$ is the **critical upper-bracketing radius** (Zhang, 2006). The decay rate of $\varepsilon_{upper,n}$ controls the consistency of the Bayesian posterior distribution (Asmuth et al., 2009). Let $\rho = \frac{1}{2}$, we have for all $t \geq 0$ and $\delta \in (0, 1)$, with probability at least $1 - \delta$,

$$\pi_n\Big(\Big\{p \in \Gamma : ||p - q||_1^2/2 \geq \frac{2\varepsilon_n + (4\eta - 2)t}{\delta/4}\Big\}\Big|X\Big) \leq \frac{1}{1 + e^{nt}}. \tag{3}$$

*Proof (sketch).* The proof is similar to that of Corollary 5.2 of Zhang (2006) (see Appendix A.5). Instead of using the critical prior-mass radius $\varepsilon_{\pi,n}$ to describe certain characteristics of the Bayesian prior, we define and use the prior-mass radius $d_\pi$, which is independent of the sample size $n$ and measures the distance between the prior and true distribution. □

Similar to BOSS, for a new MDP $m \sim M$ with hidden parameters $\omega_m$, we can define the Bayesian concentration sample complexity for the task-specific posterior: $f(s, a, \epsilon_0, \delta_0, \rho_0)$, as the minimum number $c$ such that, if $c$ IID transitions from $(s, a)$ are observed, then, with probability at least $1 - \delta_0$,

$$Pr_{m \sim posterior}(||T_m(s, a, \omega_m) - T_{m^*}(s, a, \omega_m)||_1 < \epsilon_0) \geq 1 - \rho_0. \tag{4}$$

**Lemma 2.** Assume the posterior is consistent (that is, $\varepsilon_{upper,n} = o(1)$) and set $\eta = 2$, then the Bayesian concentration sample complexity for the task-specific posterior $f(s, a, \epsilon, \delta, \rho) = O\left(\frac{d_\pi + \ln \frac{1}{\rho}}{\epsilon^2 \delta - d_\pi}\right)$.

*Proof (sketch).* This bound can be derived by directly combining Lemma 1 and Equation 4. □

The above lemma suggests an upper bound of the Bayesian concentration sample complexity using the prior-mass radius. We can further combine this result with PAC-MDP theory (Strehl et al., 2006) and derive the sample complexity of the algorithm for each new task.

**Theorem 1.** For each new task, set the sample size $K = \Theta(\frac{S^2 A}{\delta} \ln \frac{SA}{\delta})$ and the parameters $\epsilon_0 = \epsilon(1 - \gamma)^2, \delta_0 = \frac{\delta}{SA}, \rho_0 = \frac{\delta}{S^2 A^2 K}$, then, with probability at least $1 - 4\delta$, $V^{A_t}(s_t) \geq V^*(s_t) - 4\epsilon_0$ in all but $\tilde{O}(\frac{S^2 A^2 d_\pi}{\delta \epsilon^3 (1-\gamma)^6})$ steps, where $\tilde{O}(\cdot)$ suppresses logarithmic dependence.

*Proof (sketch).* The proof is based on the PAC-MDP theorem (Strehl et al., 2009) combined with the new bound for the Bayesian concentration sample complexity we derived in Lemma 2. In general, we use the same process in BOSS to verify the three required properties of PAC-MDP: optimism, accuracy and learning complexity. For each new task, the main difference between BLRL and BOSS is that we use the world-model posterior to initialize the task-specific posterior, which results in a new sample complexity bound based on the prior-mass radius. □

The result formalizes the intuition that, if we put a larger prior mass at a density that is close to the true $q$ such that $d_\pi$ is small, the sample complexity of our algorithm will be lower. In the meantime, the sample complexity is bounded by polynomial functions of the relevant quantities, showing that our training strategy preserves the properties required by PAC-MDP algorithms (Strehl et al., 2009).

## 4 SCALING UP: VARIATIONAL BAYESIAN LIFELONG RL

Directly computing the exact posterior is typically not possible for large scale problems. Instead, we propose a practical approximate algorithm, VBLRL, that uses neural networks and variational inference (Hinton & van Camp, 1993a). We model the posterior via the transition dynamics using $p(s_{t+1}, r_t|s_t, a_t; \theta), \theta \in \Theta$. The posterior, given a new state–action pair, can be rewritten via Bayes' rule:

$$p(\theta|D_t, a_t, s_{t+1}, r_t) = \frac{p(\theta|D_t)p(s_{t+1}, r_t|D_t, a_t; \theta)}{p(s_{t+1}, r_t|D_t, a_t)}, \quad (5)$$

where $D_t$ is the agent's history with all the experienced tasks up until time step $t$. As representing the posterior $p(\theta|D)$ is intractable, we approximate it through an alternative distribution $q(\theta; \phi)$ by minimizing $D_{KL}[q(\theta; \phi)||p(\theta|D)]$, leveraging variational lower bounds (Hinton & van Camp, 1993b; Houthooft et al., 2016).

We choose Bayesian neural networks (BNN) to approximate the posterior. The intuition is that, in the context of stochastic outputs, BNNs naturally approximate the hierarchical Bayesian model since they also maintain a learnable distribution over their weights and biases (Graves, 2011; Houthooft et al., 2016). We expect the uncertainty embedded in the weights and biases of networks can capture the epistemic uncertainty introduced by hidden parameters of different tasks, while we also set the outputs of the neural networks to be stochastic to capture the aleatoric uncertainty within each specific task. In our case, the BNN weights and biases distribution $q(\theta; \phi)$ can be modeled as fully factorized Gaussian distributions (Blundell et al., 2015):

$$q(\theta; \phi) = \prod_{i=1}^{|\Theta|} \mathcal{N}(\theta_i|\mu_i, \sigma_i^2), \quad (6)$$

where $\phi = \{\mu, \sigma\}$, and $\mu$ is the Gaussian's mean vector while $\sigma$ is the covariance matrix diagonal. Then, the posterior distribution over the model parameters can be computed through:

$$\phi_t = \arg\min_\phi \left[ D_{KL}[q(\theta; \phi)||p(\theta)] - \mathbb{E}_{\theta \sim q(\cdot; \phi)}[\log p(s_{t+1}, r_t|D_t, a_t; \theta)] \right], \quad (7)$$

where $p(\theta)$ represents the fixed prior distribution of $\theta$. The second term on the right hand side can be approximated through $\frac{1}{N} \sum_{i=1}^{N} \log p(s_{t+1}, r_t | D_t, a_t; \theta_i)$ with N samples from $\theta_i \sim q(\phi)$. This optimization can be performed in parallel for each $s$, keeping $\phi_{t-1}$ fixed.

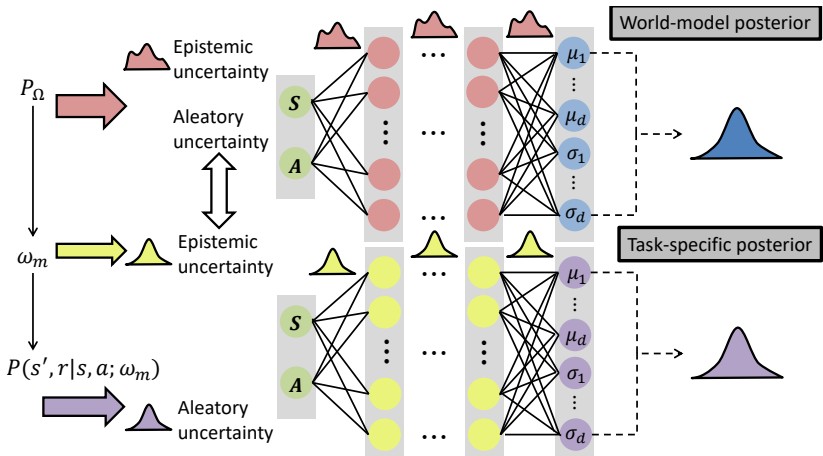

Figure 2: How VBLRL estimates different kinds of uncertainties in HiP-MDP. The world-model posterior captures the epistemic uncertainty of the general knowledge distribution (shared across all tasks controlled by the hidden parameters) via the internal variance of world-model BNN. As the learner is exposed to more and more tasks, the posterior should converge to $P_\Omega$. The task-specific posterior captures the epistemic uncertainty of the current task $m$, which comes from the alleatory uncertainty of the world model when generating $\omega_m$ for a new task, via the internal variance of task-specific BNN. The posterior should output the highest probability for $\omega$ near the true $\omega_m$ as the agent collects enough data from the task. The aleatory uncertainty of the final prediction is measured by the output variance of the prediction.

We provide the intuition of how our design capture the uncertainties of lifelong RL in Figure 2 and summarize the method in Algorithm 1. The left side of the figure shows the process of how transitions are generated from the environment's true distribution, while the other parts show how our models generate transition predictions and how they separately estimate different uncertainties generating from approximating the true underlying distribution. We employ our posterior knowledge models in the context of a model-based RL method. When encountering a new task, VBLRL first uses the model parameters (that is, $\{\mu, \sigma\}$ of weights and biases of BNN) from the general knowledge model to initialize the task-specific posterior network. The task-specific model outputs the predicted next state and reward given a state–action pair. Then, we use model-predictive control (Garcia et al., 1989) to select actions based on the generated transitions.

For planning, at each step, we begin by creating $P$ particles from the current state $s_{\tau=t}^p = s_t \forall p$. Then we sample $N$ candidate action sequences $a_{t:t+T}$ from a learnable distribution. We propagate the state-action pairs using the learned task-specific model $p_{m_i}(\cdot | s, a)$ (BNN) and use the cross entropy method (Botev et al., 2013) to update the sampling distribution to make the sampled action sequences close to previous action sequences that achieved high reward. We further calculate the cumulative reward estimated (via the learned model) for previously sampled sequences and use the mean of that distribution to select the current action.

The task-specific posterior is updated using the data collected from only the current task. The world-model posterior is updated after a few more steps with the collected transitions from **all** the visited tasks. The intuition is to guide the two posteriors to separately learn two categories of uncertainty within lifelong learning tasks. Note that other CEM-based model-based RL algorithms like PETS (Chua et al., 2018) usually maintain a set of neural networks using the same training data, and sample action sequences from each of the neural nets to achieve randomness in transitions. Besides the problem of requiring special training tricks, it is unrealistic to maintain ($\geq 30$) models for each task in lifelong RL settings. Our usage of BNNs avoids such problems as we only have to train one neural network using the same data for each task, and we can sample an unlimited number of different action sequences to cover more possibilities as needed. In PETS, the epistemic uncertainty

is estimated via the variance of the output mean of different neural networks, while in VBLRL, it is estimated via the variance of the weights and biases distribution of BNN during training. This is implied as the objective function we use to update $\phi_t$ is from minimizing $D_{KL}[q(\theta;\phi)||p(\theta|D)]$.

---

**Algorithm 1:** Variational Bayesian Lifelong RL

---

Initialize general knowledge model $p_{wm}(\cdot|s,a;\theta_{wm})$, replay buffer $D_{m_1},\cdots,D_{m_M}$
**for** *each task $m_i$ from $i = 1,2,3,\cdots,M$* **do**
    Initialize task-specific model $p_{m_i}(\cdot|s,a;\theta_{m_i})$ with general knowledge model $p_{wm}$
    **for** *each episode* **do**
        **for** *Time $t = 0$ to TaskHorizon* **do**
            Sample Actions $a_{t:t+T} \sim \text{CEM}(\cdot)$
            Propagate state particles $s_\tau^p$ with $p_{m_i}(s'|s,a)$
            Evaluate actions as $\sum_{\tau=t}^{t+T}\frac{1}{P}\sum_{p=1}^{P}p_{m_i}(r|s,a)$
            Update CEM$(\cdot)$ distribution.
            Execute optimal actions $a_{t:t+T}^*$
        **end**
        Add transitions to replay buffer $D_{m_i}$
        Update task-specific model according to Equation (7) given $D_{m_i}$
        Update general knowledge model according to Equation (7) given $\{D_{m_1},\cdots,D_{m_i}\}$
    **end**
**end**

---

## 4.1 BACKWARD TRANSFER OF VARIATIONAL BAYESIAN LIFELONG RL

In our lifelong RL setting, the agent interacts with each task for only a limited number of episodes and the task-specific model stops learning when the next task is initiated. As a result, there may exist portions of the transition dynamics in which model uncertainty remains high. However, as the world-model posterior continues to train on new tasks, it gathers more experience in the whole state space and can provide improvements in its guesses concerning the "unknown" transition dynamics, even for previously encountered tasks.

Intuitively, the performance of an agent on one task has the potential to be further improved (positive backward transfer) if there exists a sufficiently large set of state–action transition pairs of which the task-specific model's predictions are not confident due to lack of data. This type of model uncertainty is sometimes called epistemic uncertainty (Kiureghian & Ditlevsen, 2009; Ciosek et al., 2020). In our algorithm, the aleatory variability (irreducible chance in the outcome) is measured by the output variance of the prediction $\{\sigma_{r_\tau^p},\sigma_{s_\tau^p}\}$, and the epistemic uncertainty (due to lack of experience) corresponds to the uncertainty of the output mean and variance (see Definition 1 below). Thus, a straightforward method to improve a previously learned task-specific model is to find the predictions it needs to make that have high epistemic uncertainty, and replace them with the predictions from the world-model posterior, which has lower epistemic uncertainty. If we only consider reward prediction, the conditions for measuring whether a task-specific model is sufficiently confident are as follows.

**Definition 1.** Assume there exist known constants $\delta_{\mu_r},\delta_{\sigma_r}$. For a given state–action pair $(s,a)$, the task-specific model (reward) is proclaimed confident when the following conditions are satisfied:

$$\frac{\sum_{p=1}^{P}(\mu_{r_\tau^p}-\overline{\mu}_{r_\tau^p})^2}{P-1} < \delta_{\mu_r}, \quad \frac{\sum_{p=1}^{P}(\sigma_{r_\tau^p}-\overline{\sigma}_{r_\tau^p})^2}{P-1} < \delta_{\sigma_r}, \tag{8}$$

where $P$ is the number of particles. Similar definition applies to the task-specific model's next-state prediction. Intuitively, $\delta_{\mu_r}$ and $\delta_{\sigma_r}$ function as the threshold to judge whether the uncertainty of the output mean or variance for each dynamic prediction is too high to be called as a confident prediction.

The detailed backward transfer testing algorithm can be found in the appendix. In practice, it is often hard to find specific confidence thresholds $(\delta_{\mu_s},\delta_{\mu_r},\delta_{\sigma_s},\delta_{\sigma_r})$ that are effective. Instead, we implement a simpler approach: During planning, for each prediction, we compare the uncertainty of the output mean and variance of the world model and the task-specific model, and then choose the one with lower values, which indicates higher confidence level.

## 5 EXPERIMENTS

### 5.1 GRID-WORLD ITEM SEARCHING

We first evaluate BLRL in a simple Grid-World domain. Our testbed consists of a collection of houses, each of which has four rooms. The goal of each task is to find a specific object (blue, green or purple) in the current house. The type of each room is sampled based on an underlying distribution given by the environment. Each room type has a corresponding probability distribution of which kind of objects can be found in rooms of this type. Different tasks/houses vary in terms of which rooms are which types and precisely where objects are located in the room (the task's hidden parameters).

To simplify the problem, instead of modeling the whole MDP distribution, we use BLRL to model the object distribution as the Bayesian posterior and sample MDPs from the distribution. We use BOSS with a fixed prior (no intertask transfer) as our baseline. The average training performance of all 300 tasks are shown in Figure 3 top right. Each task consists of 10 epochs, with 21 sample steps for each epoch. Within the limited steps allotted for each task, BLRL is able to discover and transfer the common knowledge and helps the agent quickly adapt to new tasks as the training goes on. In comparison, running BOSS with a fixed prior is able to find the optimal policy eventually but needs more sample steps and learns more slowly than BLRL.

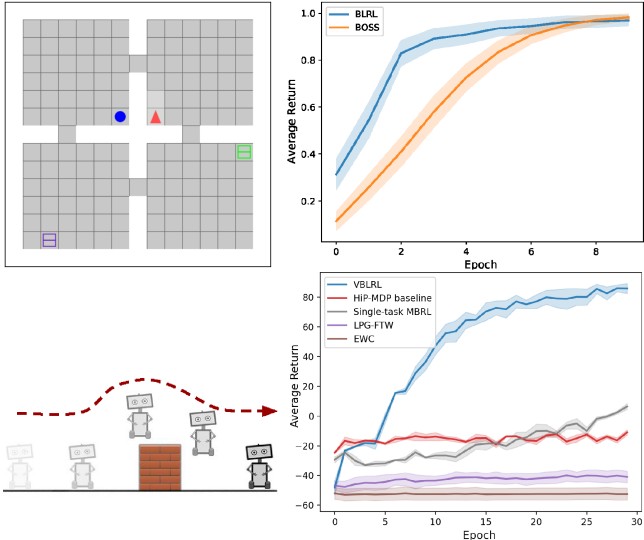

Figure 3: Top-left: Grid-World Item Searching; Top-right: Grid-World Item Searching evaluation results; Bottom-left: Box-jumping Task; Bottom-right: Box-jumping Task evaluation results.

### 5.2 BOX-JUMPING TASK

We use a simplified version of the jumping task (des Combes et al., 2018) as a testbed for the proposed algorithm VBLRL. As shown in Figure 3 bottom left, the goal of the agent is to reach the right side of the screen by jumping over the obstacle. The agent can only choose from two actions: jump and right. It will hit the obstacle unless the jump action is chosen at precisely the right time. We set different obstacle positions as different tasks, constituting the HiP-MDP hidden parameters. The 4-element state vector describes the $(x, y)$ coordinates of the agent's current position, and its velocity in the $x$ and $y$ directions.

Figure 3 bottom right presents the average performance during training across all 300 tasks. Each task is run for 30 episodes. We compare VBLRL against state-of-the-art lifelong RL methods LPG-FTW (Mendez et al., 2020) from the multi-model category, EWC (Kirkpatrick et al., 2016) from the single-model category, singel-task model-base RL baseline (using the same BNN structure and planning procedures) as well as a HiP-MDP baseline (Killian et al., 2017). For a fair comparison, we further replace the DDQN algorithm (van Hasselt et al., 2016) used in Killian et al.'s paper with CEM planning, and let the transition model also predict the reward for each state–action pair. This modified

baseline is similar to the single-model version of VBLRL (i.e., only using the world-model posterior). VBLRL clearly learns faster than the HiP-MDP baseline and reaches better final performance. Thus, in the lifelong RL setting, separating the updating processes of the world-model posterior and the task-specific posterior can lead to better learning efficiency.

## 5.3 OpenAI Gym MuJoCo Domains

We evaluate the performance of VBLRL on HiP-MDP versions of several continuous control tasks from the Mujoco physics simulator (Todorov et al., 2012), *HalfCheetah-gravity, HalfCheetah-bodyparts, Hopper-gravity, Hopper-bodyparts, Walker-gravity, Walker-bodyparts*, all of which are lifelong-RL benchmarks used in prior work[1] (Mendez et al., 2020). For each of six different domains, the task-specific hidden parameters correspond to different gravity values or different sizes and masses of the simulated body parts. More details can be found in the appendix. Compared with prior work, we substantially reduced the number of iterations that the agent can sample and train on (100 iterations for each task and a horizon of 100/200 for each iteration). We used such settings to increase the difficulty of lifelong learning and ensure that no learning-from-scratch algorithm could obtain a good policy in the available training time.

| | VBLRL | HiP-MDP baseline | LPG-FTW | EWC | Single-task MBRL |
|---|---|---|---|---|---|
| *CG*-Start | **160.68 ± 48.80** | 126.95 ± 31.41 | −81.59 ± 9.18 | −3426.76 ± 827.99 | −83.96 ± 60.10 |
| *CG*-Train | **226.72 ± 26.53** | 170.20 ± 39.92 | −29.49 ± 11.03 | −3440.66 ± 1007.50 | −40.47 ± 10.68 |
| *CG*-Back | **231.79 ± 23.49** | 97.84 ± 22.04 | −29.95 ± 11.64 | −6672.33 ± 3748.63 | / |
| *CB*-Start | **110.74 ± 41.96** | 78.95 ± 18.43 | −263.94 ± 40.80 | −5016.93 ± 1708.10 | −101.02 ± 39.11 |
| *CB*-Train | **173.97 ± 78.26** | 87.2 ± 9.42 | −217.86 ± 42.82 | −5454.52 ± 2145.82 | −58.93 ± 33.24 |
| *CB*-Back | **181.60 ± 67.50** | 116.03 ± 17.35 | −116.41 ± 65.64 | −13889.31 ± 6851.05 | / |
| *HG*-Start | −149.79 ± 28.7 | −130.54 ± 14.86 | **16.15 ± 22.83** | −614.80 ± 600.14 | −408.46 ± 36.10 |
| *HG*-Train | −125.40 ± 9.17 | −110.19 ± 22.03 | **22.98 ± 34.34** | −749.49 ± 654.04 | −386.77 ± 65.77 |
| *HG*-Back | **−55.95 ± 74.32** | −142.82 ± 16.30 | −252.36 ± 106.91 | −5816.74 ± 4103.66 | / |
| *HB*-Start | −119.29 ± 17.15 | −73.41 ± 36.99 | **−15.56 ± 48.63** | −4701.72 ± 1527.74 | −402.03 ± 30.03 |
| *HB*-Train | −99.00 ± 8.05 | −96.68 ± 38.725 | **41.29 ± 12.19** | −7384.54 ± 3232.86 | −394.77 ± 25.77 |
| *HB*-Back | **−76.47 ± 13.27** | −92.45 ± 41.50 | −186.08 ± 151.82 | −7921.96 ± 1147.94 | / |
| *WG*-Start | **−19.53 ± 4.76** | −20.86 ± 5.37 | −290.35 ± 70.95 | −467.54 ± 249.19 | −440.89 ± 59.41 |
| *WG*-Train | **7.77 ± 6.38** | 1.57 ± 2.97 | −94.34 ± 61.36 | −361.65 ± 260.76 | −359.82 ± 45.44 |
| *WG*-Back | **18.56 ± 7.42** | 9.79 ± 6.32 | −90.41 ± 64.66 | −734.90 ± 413.40 | / |
| *WB*-Start | **−43.01 ± 10.82** | −64.62 ± 28.02 | −315.60 ± 31.66 | −1140.62 ± 180.21 | −437.96 ± 26.24 |
| *WB*-Train | **−2.66 ± 3.67** | −31.29 ± 30.07 | −187.98 ± 72.03 | −1131.03 ± 451.38 | −367.79 ± 74.37 |
| *WB*-Back | **4.04 ± 2.95** | −33.75 ± 40.00 | −66.97 ± 74.71 | -−2563.70 ± 692.70 | / |

Table 1: Results on OpenAI Gym Mujoco domains. *CG* denotes **Cheetah-Gravity**, *CB* denotes **Cheetah-Bodyparts**, *HG* denotes **Hopper-Gravity**, *HB* denotes **Hopper-Bodyparts**, *WG* denotes **Walker-Gravity**, *WB* denotes **Walker-Bodyparts**.

The results are shown in Table 1. We compare our algorithm against the three algorithms described in Section 5.2. For all six domains, we report the average performance of all the tasks at the beginning of training (**Start**) and after all training for each new task (**Train**), as well as the average performance for all previous tasks after training for a given number of tasks, which is the backward transfer test (**Back**). As shown in the results, our method VBLRL shows better performance on all three test stages of the HalfCheetah domain and Walker domain, as well as better backward transfer performance on Hopper-gravity and Hopper-bodyparts than the other three algorithms. LPG-FTW exhibits better forward training performance in the Hopper domain, but still shows some signs of catastrophic forgetting as there is a huge gap between its training performance and backward performance. EWC fails in most of the tasks as the tasks are diverse and we set very limited sample steps for each task, which means it is hard to directly learn a single shared policy that achieves good performance. The HiP-MDP baseline shows good results on some of the tasks because it is more sample-efficient to learn a shared model across all the tasks and easier to capture the world-model uncertainty. However, it cannot achieve as good performance as VBLRL as it is hard to model the task-specific uncertainty using only one model across all tasks, which also leads to the negative backward transfer performance on Cheetah-Gravity. Comparing VBLRL's performance on the **Train** stage and **Back** stage, we also find that it shows positive backward transfer results on most of the tasks, without showing patterns of catastrophic forgetting. Overall, VBLRL's world-model posterior contributes to better forward

---

[1]We changed the environment settings for Hopper and Walker to make them amenable to model-based RL following Wang et al. (2019).

transfer performance (**Start**), the learning of task-specific posterior contributes to better forward transfer training for each new task (**Train**), and the combination of these two posteriors guides the agent to achieve better backward transfer performance (**Back**).

## 6 RELATED WORK

HiP-MDPs (Doshi-Velez & Konidaris, 2016) provide a framework for studying lifelong RL. Published HiP-MDP methods use Gaussian Processes (Doshi-Velez & Konidaris, 2016) or Bayesian neural networks (Killian et al., 2017) to find a single model that works for all tasks, which may trigger *catastrophic forgetting*. Other single-model lifelong RL algorithms encourage transfer across tasks by modifying objective functions. EWC (Kirkpatrick et al., 2016) imposes a quadratic penalty that pulls each weight back towards its old values by an amount proportional to its importance for performance on previously-learned tasks to avoid forgetting. There are several extensions of this work based on the core idea of modifying the form of the penalty (Li & Hoiem, 2017; Zenke et al., 2017; Nguyen et al., 2018). Another category of lifelong RL methods uses multiple models with shared parameters and task-specific parameters to avoid or alleviate the catastrophic problem (Bou-Ammar et al., 2014; Isele et al., 2016a; Mendez et al., 2020). The drawback of this method is that it is hard to incorporate the knowledge learned from previous tasks during initial training on a new task (Mendez et al., 2020).Nagabandi et al. (2019) introduce a model-based continual learning framework based on MAML, but they focus on discovering when new tasks were encountered without access to task indicators.

Research in Meta-RL (Wang et al., 2016; Finn et al., 2017) and multi-task RL (Parisotto et al., 2016; Teh et al., 2017) settings also attempts to find ways to accelerate learning by transferring knowledge from different tasks. Some work employs the MAML framework with Bayesian methods to learn a stochastic distribution over initial parameter (Yoon et al., 2018; Grant et al., 2018; Finn et al., 2018). Other work uses the collected trajectories to infer the hidden parameter, which is taken as an additional input when computing the policy (Rakelly et al., 2019; Zintgraf et al., 2020; Fu et al., 2021). Our method, however, focuses on problems where the tasks arrive sequentially instead of having a large number of tasks available at the beginning of training. This sequential setting makes it hard to accurately infer the hidden parameters, but opens the door for algorithms that support backward transfer. Further, our method approximates the true HiP-MDP model by learning the Bayesian posterior over past tasks and uses it to initialize a model for each new task, encouraging the agent to explore places where the epistemic uncertainty of the world model is high.

Some prior work uses Bayesian methods in RL to quantify uncertainty over initial MDP models (Ghavamzadeh et al., 2015; Asmuth & Littman, 2011; Guez et al., 2012). Several algorithms start from the idea of sampling from a posterior over MDPs for Bayesian RL, maintaining Bayesian posteriors and sampling one complete MDP (Strens, 2000; Wilson et al., 2007a) or multiple MDPs from this distribution (Asmuth et al., 2009). Instead of focusing on single-task RL, our algorithm aims to find a posterior over the common structure among multiple tasks. Wilson et al. (2007a) uses a hierarchical Bayesian infinite mixture model to learn a strong prior that allows the agent to rapidly infer the characteristics of new environment based on previous tasks. However, it only infers the category label of a new MDP and uses that information to find parameter values. Moreover, their method only works in discrete settings and cannot be applied to the kind of continuous problems we included in our evaluation. Lifelong learning has also been widely studied within the supervised learning domain with explicit performance bounds (Baxter, 2000; Pentina & Lampert, 2014). Our work is one of the first papers to give explicit sample complexity bounds for lifelong reinforcement learning algorithm, where data efficiency is essentially important.

## 7 CONCLUSION

To address the lifelong RL problems, our work proposed to distill the shared knowledge from similar MDPs and maintain a Bayesian posterior to approximate the distribution derived from that knowledge. We gave a sample complexity analysis of the algorithm in the finite MDP setting. Then, we extended our method to use variational inference, which scales better and supports both backward and forward transfer. Our experimental results show that the proposed algorithms enables faster training on new tasks through collecting and transferring the knowledge learned from preceding tasks.

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

## A APPENDIX

### A.1 BACKWARD TRANSFER

We take backward transfer to be the influence of subsequent learning on the agent's performance of a previous task. Negative backward transfer, which is also known as *catastrophic forgetting*, describes the scenario in which the agent's progress on a task is erased after learning new tasks (French, 1993; Carpenter & Grossberg, 1988; Robins, 1995). Our algorithm avoids catastrophic forgetting as it maintains a separate task-specific model for each task. In the mean time, there are circumstances in which VBLRL exhibits positive backwards transfer as introduced in the main text.

---

**Algorithm 2:** Variational Bayesian Lifelong RL (Backward transfer)

---

**input :** Test task $m_i$, model uncertainty thresholds $\delta_{\mu_s}, \delta_{\mu_r}, \delta_{\sigma_s}, \delta_{\sigma_r}$
**for** *Time $t = 0$ to TaskHorizon* **do**
    **for** *Trial $k = 1$ to $K$* **do**
        Sample Actions $a_{t:t+T} \sim \text{CEM}(\cdot)$
        **for** *each action* **do**
            Propagate state particles $s_\tau^p$ and reward $r_\tau^p$ with $T_{m_i}(s', r|s, a)$
            **if** $T_{m_i}(s', r|s, a)$ *is not confident (**Definition 1**)* **then**
                Repropagate state particles $s_\tau^p$ and reward $r_\tau^p$ with $T_e(s', r|s, a)$
            **end**
        **end**
        Evaluate actions as $\sum_{\tau=t}^{t+T} \frac{1}{P} \sum_{p=1}^{P} T_{m_i}(r|s, a)$
        Update CEM($\cdot$) distribution.
    **end**
    Execute optimal actions $a_{t:t+T}^*$
**end**

---

The difference between the backward transfer version and the forward training version is that, when we do predictions for the state particles, each time we will also calculate the confidence level (Definition 1) of the task-specific model for this particular transition. If the task-specific model is not confident enough for this state–action pair, we will use the world-model instead and recalculate the prediction value.

In practice, it is often hard to find specific confidence thresholds $(\delta_{\mu_s}, \delta_{\mu_r}, \delta_{\sigma_s}, \delta_{\sigma_r})$ that are effective. Instead, we implement a simpler approach: During planning, for each prediction, we compare the uncertainty of the output mean and variance of the world model and the task-specific model, and then choose the one with lower values, which indicates higher confidence level. Here is a link to the learned behaviors of VBLRL `https://youtu.be/I7RAT6g9v5w`. The video shows the agent trained by our algorithm learns some meaningful behaviors at the current setting within limited steps, and performs better compared with the behaviors learned by other baseline algorithms.

### A.2 BNN MODEL

As we model the transition models as Gaussian distributions:

$$T(\cdot|s, a) = \mathcal{N}(f_\theta^\mu(s, a), f_\theta^\sigma(s, a)) \tag{9}$$

The function $f_\theta$ represented as a Bayesian neural network parameterized by $\theta$, which is further modeled as the posterior distribution parameterized by $\phi$, predicts the mean $\mu_s, \mu_r$ and variance $\sigma_s, \sigma_r$ given current state and action $s, a$. To make it clearer, we can view the BNN model in VBLRL as an infinite neural network ensemble by integrating out its parameters:

$$T(s', r|s, a) = \int_\Theta T(s', r|s, a; \theta) q(\theta; \phi) d\theta \tag{10}$$

Compared to previous model-based algorithms that use finite number of neural network ensembles (e.g. PETS), our choice of BNN is more suitable for lifelong RL as we only need to maintain one neural network for each task, and we can sample an unlimited number of predictions from it which better estimates the uncertainty and is essential in our setting where both dynamic function and reward function are not given unlike prior model-based RL methods.

## A.3 BLRL ALGORITHM

---

**Algorithm 3:** Lifelong Bayesian Sampling Approach Algorithm

---

**input :** $K$, $B$
initialize MDP set, the world-model posterior $q_e(s_{t+1}, r_t | s_t, a_t)$;
**for** *each MDP $m_i$* **do**
    $N_{s,a} \leftarrow 0, \forall s, a$
    $do\_sample \leftarrow$ TRUE ;
    initialize the task-specific posterior $q_\theta^{m_i}(s_{t+1}, r_t | s_t, a_t) \leftarrow q_e(s_{t+1}, r_t | s_t, a_t)$;
    **for** *all timesteps $t = 1, 2, 3, \ldots$* **do**
        **if** *$do\_sample$* **then**
            Sample $K$ models $m_{i_1}, m_{i_2}, \cdots, m_{i_K}$ from the task-specific posterior
                $q_\theta^{m_i}(s_{t+1}, r_t | s_t, a_t)$.;
            Merge the models into the mixed MDP $m_i^{\#}$;
            Solve $m_i^{\#}$ to obtain $\pi^*_{m_i^{\#}}$ ;
            $do\_sample \leftarrow$ FALSE
        **end**
        Use $\pi^*_{m_i^{\#}}$ for action selection: $a_t \leftarrow \pi_{m_i^{\#}}(s_t)$ and observe reward $r_t$ and next state $s_{t+1}$ ;
        $N_{s_t,a_t} \leftarrow N_{s_t,a_t} + 1$;
        Update the task-specific posterior distribution $q_\theta^{m_i}(s_{t+1}, r_t | s_t, a_t)$ for the current MDP;
        **if** *$N_{s_t,a_t} = B$* **then**
            Update the world-model posterior distribution $q_e(s_{t+1}, r_t | s_t, a_t)$ with the collected
              transitions;
            $do\_sample \leftarrow$ TRUE
        **end**
    **end**
**end**

---

## A.4 EXPERIMENTAL SETTING

### A.4.1 GRID-WORLD ITEM SEARCHING

Our testbed consists of a collection of houses, each of which has four rooms. The goal of each task is to find a specific object (blue, green or purple) in the current house. The type of each room is sampled based on an underlying distribution given by the environment. Each room type has a corresponding probability distribution of which kind of objects can be found in rooms of this type. Different tasks/houses vary in terms of which rooms are which types and precisely where objects are located in the room (the task's hidden parameters). Room types are sampled from a joint distribution.

| Room type probability | **Room 1** | **Room 2** | **Room 3** | **Room 4** |
|:---:|:---:|:---:|:---:|:---:|
| Top-left | 0.4 | 0 | 0.4 | 0.2 |
| Bottom-left | 0 | 0.8 | 0 | 0.2 |
| Top-right | 0.1 | 0 | 0 | 0.9 |
| Bottom-right | 0 | 0 | 0.8 | 0.2 |

Table 2: Room type probability distribution

### A.4.2 BOX-JUMPING TASK

We use a simplified version of jumping task (des Combes et al., 2018) as a simple testbed for the proposed algorithm VBLRL. We select a random position of obstacle between $15 \sim 33$ for each task. The 4-element state vector describes the $(x, y)$ coordinates of the agent's current position, and its velocity in the $x$ and $y$ directions. The agent can choose from two actions: jump and right. The reward function for this box-jumping task is:

$$R_t = \mathbb{I}\{s_t \text{ reach the right wall}\} - \mathbb{I}\{s_{t+1} \text{ hit the obstacle}\} + \dot{x}_t \cdot \mathbb{I}\{s_{t+1} \text{ not hit the obstacle}\} \quad (11)$$

| Object type probability | Blue ball | Green box | Purple box |
|---|---|---|---|
| **Room 1** | 0 | 0.3 | 0 |
| **Room 2** | 0 | 0.2 | 1 |
| **Room 3** | 0.6 | 0 | 0 |
| **Room 4** | 0 | 0 | 0 |

Table 3: Object type probability distribution

### A.4.3 OPENAI GYM MUJOCO DOMAINS

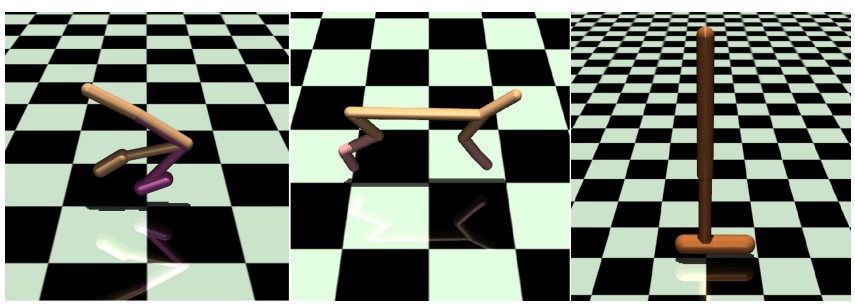

Figure 4: Left: Walker-2D; Middle: Halfcheetah; Right: Hopper

Similar to (Mendez et al., 2020), we evaluated on the HalfCheetah, Hopper, and Walker-2D environments. For the gravity domain, we select a random gravity value between $0.5g$ and $1.5g$ for each task. For the body-parts domain, we set the size and mass of each of the four parts of the body (head, torso, thigh, and leg) to a random value between $0.5\times$ and $1.5\times$ its nominal value. As shown in Appendix C of (Mendez et al., 2020), these changes lead to highly diverse tasks for lifelong RL. Further, as required by model-based deep RL methods, we change the environment settings for Hopper and Walker following (Wang et al., 2019). When implementing VBLRL, we found that in the first few

| Name of Environment | Reward function |
|---|---|
| **HalfCheetah** | $\dot{x}_t - 0.1\|\|a_t\|\|_2^2$ |
| **Hopper** | $\dot{x}_t - 0.1\|\|a_t\|\|_2^2 - 3.0 \times (z_t - 1.3)^2$ |
| **Walker2D** | $\dot{x}_t - 0.1\|\|a_t\|\|_2^2 - 3.0 \times (z_t - 1.3)^2$ |

Table 4: Reward functions for OpenAI Gym domains

episodes of each new tasks, the agent hasn't collected enough samples of the new task, which results in overfitting problems when training the task-specific. Thus, we use the world-model posterior instead to do the first few rounds of predictions and let the task-specific model begin training after collecting enough samples. The world-model has lower possibility of overfitting as its training data comes from all the previous tasks and has much larger quantity. We list the other implementation details below. The planning horizons are selected from values suggested by previous model-based RL papers (Chua et al., 2018; Wang et al., 2019).

For LPG-FTW and EWC, we use the original source code[2] with parameters and model architectures suggested in the original paper. Specifically, we select step size from $\{0.005, 0.05, 0.5\}$. For LPG-FTW, e use $\lambda = 1e - 5, \mu = 1e - 5$ and select $k$ from 3,5,10. For EWC, we select $\lambda$ from $\{1e - 6, 1e - 7, 1e - 4\}$. For HiP-MDP baseline, we modify the original algorithm for a fair comparison. We replace the DDQN algorithm used in Killian et al.'s paper with the exact same CEM planning method we used in VBLRL as well as the same parameters. And we use the same model architecture of Bayesian Neural network by modifying the baseline algorithm to also predict reward for each state-action pair (the original method only considers next-state prediction).

---

[2]https://github.com/Lifelong-ML/LPG-FTW

| Hyper-parameters | CG | CB | HG | HB | WG | WB |
|---|---|---|---|---|---|---|
| # iterations | 100 | 100 | 100 | 100 | 100 | 100 |
| # Steps (each iteration) | 100 | 100 | 200 | 200 | 200 | 200 |
| learning rate (world model) | 0.001 | 0.001 | 0.0005 | 0.0005 | 0.0005 | 0.0005 |
| learning rate (task-specific model) | 0.0005 | 0.0005 | 0.0002 | 0.0002 | 0.0002 | 0.0002 |
| planning horizon | 20 | 20 | 30 | 30 | 1 | 1 |
| kl-divergence weight | 0.0001 | 0.0001 | 0.0001 | 0.0001 | 0.0001 | 0.0001 |
| # particles (CEM) | 50 | 50 | 50 | 50 | 50 | 50 |
| batch size (world-model) | $8 \times 64$ | $8 \times 64$ | $8 \times 64$ | $8 \times 64$ | $8 \times 64$ | $8 \times 64$ |
| batch size (task-specific) | 256 | 256 | 256 | 256 | 256 | 256 |
| # tasks | 40 | 40 | 30 | 30 | 40 | 40 |
| search population size | 500 | 500 | 500 | 500 | 500 | 500 |
| # elites (CEM) | 50 | 50 | 50 | 50 | 50 | 50 |

Table 5: Room type probability distribution

For BOSS and BLRL, we set the number of sampled models $K = 5$, and $\gamma = 0.95, \Delta = 0.01$ for value iteration.

### A.5 DETAILED PROOF FOR LEMMA 1

Following the same settings introduced in section 5 of (Zhang, 2006), we define the *resolvability of standard Bayesian posterior* as:

$$
\begin{aligned}
r_n(q) &= \inf_{\omega}[\mathbb{E}_\pi \omega(\theta) D_{KL}(q||p(\cdot|\theta)) + \frac{1}{n} D_{KL}(\omega d\pi || d\pi)] \\
&= -\frac{1}{n} \ln \mathbb{E}_\pi e^{-n D_{KL}(q||p(\cdot|\theta))}.
\end{aligned}
\tag{12}
$$

We refer the readers to Zhang's paper for further explanation of the denotations. Intuitively, the Bayesian resolvability controls the complexity of the density estimation process. Based on this definition and our previous definitions of $d_\pi$, we can derive a simple and intuitive estimate of the standard Bayesian resolvability.

**Lemma 3.** The *resolvability of standard Bayesian posterior* defined in (12) can be bounded as

$$
r_n(q) \leq \frac{n+1}{n} d_\pi
$$

*proof.* For all $d > 0$, we have

$$
\begin{aligned}
r_n(q) &= -\frac{1}{n} \ln \mathbb{E}_\pi e^{-n D_{KL}(q||p(\cdot|\theta))} \leq -\frac{1}{n} \ln[e^{-nd} \pi(p \in \Gamma : D_{KL}(q||p) \leq d)] \\
&= d + \frac{1}{n} \times [-\ln \pi(p \in \Gamma : D_{KL}(q||p) \leq d)] \leq \frac{n+1}{n} d_\pi
\end{aligned}
$$

$\square$

This bound links the Bayesian resolvability to the number of samples $n$ and prior-mass radius $d_\pi$ which is a fixed property of the density given a specific prior and the true underlying density. Intuitively, the Bayesian posterior is better behaved when the Bayesian prior is closer to the true distribution ($d_\pi$ is smaller) and more samples are used (n is larger).

Now we can prove the main theorem of Lemma 1. Let $\rho = \frac{1}{2}$, $\epsilon_t = \frac{2\varepsilon_n + (4\eta - 2)t}{\delta/4}$, define $\Gamma_1 = \{p \in \Gamma : D_\rho^{Re}(q||p) < \epsilon_t\}$ and $\Gamma_2 = \{p \in : D_\rho^{Re}(q||p) \geq \epsilon_t\}$. We let $a = e^{-nt}$ and define $\pi'(\theta) = a\pi(\theta)C$ when $\theta \in \Gamma_1$ and $\pi'(\theta)C$ when $\theta \in \Gamma_2$, where the normalization constant $C = (a\pi(\Gamma_1) + \pi(\Gamma_2))^{-1} \in [1, 1/a]$. Firstly,

$$
\mathbb{E}_X \pi'(\Gamma_2|X)\epsilon_t \leq \mathbb{E}_X \mathbb{E}_{\pi'} \pi'(\theta|X) \frac{1}{2}||p - q||_1^2 \leq \mathbb{E}_X \mathbb{E}_{\pi'} \pi'(\theta|X) D_{KL}(q||p)
$$

according to the Markov inequality (with probability at least $1 - \delta$) and Pinsker's inequality. Then according to Theorem 5.2 and Proposition 5.2 in Zhang's paper,

$$
\mathbb{E}_X \mathbb{E}_{\pi'} \pi'(\theta|X) D_{KL}(q||p) \leq \frac{\eta \ln \mathbb{E}_{\pi'} e^{-n D_{KL}(q||p(\cdot|\omega))}}{\rho(\rho - 1)n}
$$

$$
+ \frac{\eta - \rho}{\rho(1 - \rho)n} \inf_{\{\Gamma_j\}} \ln \sum_j \pi'(\Gamma_j)^{(\eta-1)/(\eta-\rho)} (1 + r_{ub}(\Gamma_j))^n
$$

$$
\leq \frac{\eta t - (\eta/n) \ln \mathbb{E}_\pi e^{-n D_{KL}(q||p(\cdot|\omega))}}{\rho(1 - \rho)} + \frac{\eta - \rho}{\rho(1 - \rho)} \left[ \frac{(\eta - 1)t}{\eta - \rho} + \varepsilon_{upper,n} \left( \frac{\eta - 1}{\eta - \rho} \right) \right]
$$

$$
= \frac{(2\eta - 1)t}{\rho(1 - \rho)} + \frac{-(\eta/n) \ln \mathbb{E}_\pi e^{-n D_{KL}(q||p(\cdot|\omega))} + (\eta - \rho)\varepsilon_{upper,n}((\eta - 1)/(\eta - \rho))}{\rho(1 - \rho)}
$$

Then, using the definitions of $d_\pi$, we further obtain

$$
\mathbb{E}_X \pi'(\Gamma_2|X) \epsilon_t
$$

$$
\leq \frac{(2\eta - 1)t}{\rho(1 - \rho)} + \frac{\eta \inf_{d>0}[d - \frac{1}{n} \ln \pi(\{p \in \Gamma : D_{KL}(q||p) \leq d\})] + (\eta - \rho)\varepsilon_{upper,n}((\eta - 1)/(\eta - \rho))}{\rho(1 - \rho)}
$$

$$
\leq \frac{(2\eta - 1)t}{\rho(1 - \rho)} + \frac{\eta(1 + \frac{1}{n})d_\pi + (\eta - \rho)\varepsilon_{upper,n}((\eta - 1)/(\eta - \rho))}{\rho(1 - \rho)}
$$

$$
= \frac{(2\eta - 1)t + \varepsilon_n}{\rho(1 - \rho)}
$$

We use $\eta$ instead of $\gamma$ which is used in the original paper to avoid confusion with the discount factor. Then we further divide both sides by $\epsilon_t$ and obtain $\pi'(\Gamma|X) \leq 0.5$. Then by definition,

$$
\pi(\Gamma_2|X) = a\pi'(\Gamma_2|X)/(1 - (1-a)\pi'(\Gamma|X))
$$

$$
\leq \frac{a}{a+1} = \frac{1}{1 + e^{nt}}
$$

Thus, we get the desired bound.

### A.6 Detailed proof for Theorem 1

**Theorem 2.** (Full version of the bound in Theorem 1) For each new task, set the sample size $K = \Theta(\frac{S^2 A}{\delta} \ln \frac{SA}{\delta})$ and the parameters $\epsilon_0 = \epsilon(1 - \gamma)^2, \delta_0 = \frac{\delta}{SA}, \rho_0 = \frac{\delta}{S^2 A^2 K}$, then, with probability at least $1 - 4\delta$, $V^{A_t}(s_t) \geq V^*(s_t) - 4\epsilon$ in all but $O(\frac{S^2 A^2 (d_\pi + \ln \frac{S^2 A^2 K}{\delta})}{\delta \epsilon^3 (1 - \gamma)^6} \ln \frac{1}{\delta} \ln \frac{1}{\epsilon(1 - \gamma)})$ steps.

First, we would like to introduce two lemma from BOSS (Asmuth et al., 2009):

**Lemma 4.** The sample size $K = \Theta(\frac{S^2 A}{\delta} \ln \frac{SA}{\delta})$ suffices to guarantee $V_m^*(s) \geq V^*(s)$ for all $s$ during the entire learning process with probability at least $1 - \delta$.

**Lemma 5.** If the knownness parameter $B = max_{s,a} f(s, a, \epsilon, \frac{\delta}{SA}, \frac{\rho}{S^2 A^2 K})$, then the transition function of all the sampled models are $\epsilon$-close (in the $l_1$ sense) to the true transition function for all the known state-action pairs during the entire learning process with probability at least $1 - \delta - \rho$.

*proof.* Given discrete state and action spaces, the proof that BLRL on each new task is PAC-MDP depends on three main assumptions, following a general PAC-MDP theorem from (Strehl et al., 2006):

1. Learning complexity condition. In our settings, a state-action pair is claimed to be known after being visited for $B$ times. In the meantime, there are $SA$ unknown state-action pairs in total at the beginning of each task, so the bounded discoveries condition is guaranteed.

2. Optimism. This is guaranteed by Lemma 3. We construct a new hyper-model each time a discovery event occurs. The values for each unknown state in every hyper-model are optimistic.

3. Accuracy. For each new task, since the prior $\pi$ has bounded sample complexity of $B$ for $\delta_0$ and $\epsilon_0$, a known state-action pair will be locally $\epsilon_0$-$accurate$ with probability at least $1 - \delta_0$. Given the definition of Bayesian concentration sample complexity and Lemma 4, $\epsilon_0 = \epsilon(1 - \gamma)^2$ translates into an $\epsilon$ error bound in the value function (Asmuth et al., 2009).

Finally, given Lemma 2, we know $f(s, a, \epsilon_0, \delta_0, \rho_0) = O\left(\frac{d_\pi + \ln \frac{1}{\rho_0}}{\epsilon_0^2 \delta_0 - d_\pi}\right)$, let $\epsilon_0 = \epsilon(1 - \gamma)^2, \delta_0 = \frac{\delta}{SA}, \rho_0 = \frac{\delta}{S^2 A^2 K}$, we can get $f(s, a, \epsilon_0, \delta_0, \rho_0) = O\left(\frac{SA(d_\pi + \ln \frac{S^2 A^2 K}{\delta})}{\delta \epsilon^2 (1 - \gamma)^4}\right)$. We can get the final form of the bound by replacing $B$ with these quantities.

$\square$

## A.7 ADDITIONAL EXPLANATION OF THE ALGORITHM

Here we first provide an example to help the readers better understand our plate notation. In our Gridworld Item Searching case, $\Psi$ represents the parameters of $P_\Omega$, which is the room-type and object distribution. For each task, the environment samples a hidden parameter $\omega$, which is the actual room and object layout of this house, from this distribution $P_\Omega$. The sampled $\omega$ then will result in an MDP $m$ and let the agent interact with it.

### A.7.1 PLANNING ALGORITHM

With the transition dynamics and reward functions, a planning algorithm like CEM is not the only way to solve the MDP to get an optimal policy. Another option would be using other Deep RL algorithms like Soft actor-critic (Haarnoja et al., 2018) with data generated from the model. However, in this case, incorporating a deep RL algorithm means that we need to introduce additional neural networks (that is, policy/value networks) for each task. The update signal from the RL loss is usually stochastic and weak, which is even worse in this case when our model is still far from accurate. So, here we assume applying a planning algorithm is a better way to get the policy.

## A.8 COIN EXAMPLE

Consider a coin-flipping environment. We want to find the sample complexity of the unbiased coin (i.e. How many times we need to flip this coin such that our posterior samples are accurate.). Consider a Dirichlet prior, $\alpha_0 = (n_1, n_2)$ and $\theta_0 = (\frac{1}{2}, \frac{1}{2})$. We want to find sample complexity $B$ such that the posterior likelihood for a coin with heads likelihood in $[0.5 - \epsilon, 0.5 + \epsilon]$ is at least $1 - \delta$.

Note that the Dirichlet distribution on the two-dimensional simplex is the Beta distribution. The Multinomial distribution with two outcomes is the Binomial distribution. That is, given the process

$$H \sim Bin(H|\rho = 0.5, B), \tag{13}$$

$$\hat{\rho} \sim Beta(\hat{\rho}|\alpha = H + n_1, \beta = B - H + n_2), \tag{14}$$

choose a value $B$ such that

$$P(0.5 - \epsilon \leq \hat{\rho} \leq 0.5 + \epsilon) \geq 1 - \delta, \tag{15}$$

$$\sum_{H=0}^{B} Bin(H|\rho = 0.5, B) \cdot \int_{\hat{\rho} = 0.5 - \epsilon}^{0.5 + \epsilon} Beta(\hat{\rho}|\alpha = H + n_1, \beta = B - H + n_2) \geq 1 - \delta. \tag{16}$$

Here, $n_1$ and $n_2$ capture the prior. The smallest $B$ that satisfies Equation 16 can be found numerically.

We set $\epsilon = 0.1$ and $\delta = 0.3$. Here are the results of sample complexity $B$ given different values of $n_1, n_2$:

We fix the sum of $(n_1, n_2)$ as 10. As shown in the results, the value of sample complexity $B$ becomes lower as we use a more accurate prior (from $(10, 0)$ to $(5, 5)$).

| $(n_1, n_2)$ | lowest $B$ |
|---|---|
| (0,10) | 78 |
| (1,9) | 68 |
| (2,8) | 58 |
| (3,7) | 49 |
| (4,6) | 42 |
| (5,5) | 40 |
| (6,4) | 42 |
| (7,3) | 48 |
| (8,2) | 58 |
| (9,1) | 68 |
| (10,0) | 78 |

In general, for the task-specific posterior, we can relate $B, \epsilon$ and $\delta$ with the following equation:

$$\int_{P_0} Dir(P_0 | \Phi^{True}) \Big[ \sum_{\mathbf{N}: ||\mathbf{N}||_1 = B} Mult(N | P_0, B) \Big[ \int_{P: ||P(\Phi) - P_0(\Phi)|| \leq \epsilon} Dir(P | \Phi_{old} + \mathbf{N}) dP \Big] \Big] dP_0 \geq 1 - \delta$$

(17)

For the world model posterior:

$$\sum_{\mathbf{N}: ||\mathbf{N}||_1 = B_w} Mult(N | P_{w_0}, B_w) \Big[ \int_{P: ||P_w(\Phi) - P_{w_0}(\Phi)|| \leq \epsilon} Dir(P_w | \Phi_{w_{old}} + \mathbf{N}) dP_w \Big] \geq 1 - \delta \quad (18)$$

For each task, first we pick a true model $P_0$ according to the true distribution and initialize the task-specific prior $\Phi_{old} = \Phi_w$. Then, we make some observations from the world. Once we have the true model and the observations, we can calculate how many models are $\epsilon$-close to the true model, weighted according to their posterior likelihood.

