# OpenReview forum: "Bayesian Exploration for Lifelong Reinforcement Learning"
_ICLR.cc/2022/Conference — ICLR 2022 Submitted_

### Official Review · Reviewer_7Ubo · 2021-11-01

**Correctness:** 2
**Technical Novelty And Significance:** 2
**Empirical Novelty And Significance:** 2
**Recommendation:** 3
**Confidence:** 3

**Main Review:**

The main theoretical contribution suggested in the paper is a PAC-MDP of Theorem 1 for a single task. This theorem is based on Lemma 2, for which a full proof is not provided in the main text or appendix, so its veracity cannot be verified. Moreover, it is based on the assumption that the posterior is consistent, which I believe is what needs to be shown in a meta-learning setup, and cannot be assumed. The form of the bound is also strange as it depends on \delta rather than on \ln(1/\delta) as in previous bounds (e.g., Strehl 2006 and Asmuth 2009), and its dependence on \gamma and \epsilon is also worse than previous bounds.  As far as I understand this is a bound for the single instance setting rather than for meta-learning.

Following the theoretical part the authors develop a variational approach based on probabilistic networks that model the posterior distribution. As far as I am aware their variational approach is rather standard, although the authors do not refer to previous work. Also, their use of model-predictive control is common in current ML applications, but, again, this is not mentioned or discussed (e.g., the length of the future horizon and how is it selected).

The authors conclude by presenting numerical simulations for a grid-world and some MuJoco problems for continuous control. While the method compares well to simple baselines it is hard to assess performance relative to more recent work such as Liu et al., "Taming MAML: Efficient Unbiased Meta-Reinforcement Learning" and the other baselines measured in it.


**Summary Of The Paper:**

The paper deals with the problem of lifelong RL, also referred to as meat-RL, where an agent attempts to solve a sequence of tasks in order to facilitate the solution of a novel task. The framework follows that of Baxter 2000 (albeit that paper deals with supervised learning), and has been widely studied in recent years.  The basic assumption is that the tasks are drawn from an underlying task-distribution, and each task (an MDP) is stochastically selected from  a task-specific distribution. The authors work with a Bayesian framework, assuming a hierarchical distribution of the two levels, and learn the two levels separately. This framework has the advantage of providing both estimates and uncertainty estimates. For the discrete case they present a sample complexity analysis, and suggest a variational approach to practical learning. Finally, experiments are provided supporting the utility of the approach.

The formal framework is that of hidden-parameter MDPs (HiP-MDPs) from Doshi-Velez 2016, and each MDP is modeled based on a transition and a reward model based on a hidden parameter. As more tasks are encountered the posterior over world models sharpens, and, being used to learn new tasks, is expected to facilitate learning. The learning of each new task is as in BOSS, and takes place by sampling from the learned MDP distribution, creating a mixed MDP, and using standard model-based approaches to solve these.

**Summary Of The Review:**

The paper is phrased within a sequential approach to meta-learning that has been widely studied within the supervised learning community (e.g., Baxter 2000, Pentina and Lambert, "A PAC-Bayesian Bound for Lifelong Learning" 2014 and much later work), with explicit performance bounds. It would be nice to acknowledge these roots. The present approach is plausible, and combines previous work, such as HiP-MDPs, BOSS, variational Bayes, in a sensible manner. However, I do not find that the level on innovation in this combination of approaches suffices for publication at ICLR, nor did I find the theoretical or experimental results of sufficient  interest (see comments above).

Following rebuttal: Following the authors' response and my response to their rebuttal I have lowered my assigned grade due to my dissatisfaction with their replies, which served to enhance my existing concerns about the paper.

---

> ### Author Response · Authors · 2021-11-19
> **Response to Reviewer 7Ubo**
>
> Q1: Missing reference and full proof.
>
> A: We revised the paper, and have included the references in the related work section, as well as the full proof for the theory in the appendix section A.5 & A.6. We hope the new version is clearer.
>
> Q2: “It is based on the assumption that the posterior is consistent, which I believe is what needs to be shown in a meta-learning setup, and cannot be assumed”
>
> A:  As the reviewer also mentioned, this is a bound for the single instance setting, where we initialize the agent with a learned prior and show how that affects the sample complexity of solving one single task. Thus, the assumption about the posterior in this case has nothing to do with the meta-learning setup as we independently learn a task-specific posterior for each task, which our bound is focused on. “The posterior is consistent” is the same assumption made in BOSS (Asmuth et al., 2009), and is reasonable for such single-task cases.
>
> Q3: “The form of the bound is also strange as it depends on \delta rather than on \ln(1/\delta) as in previous bounds (e.g., Strehl 2006 and Asmuth 2009), and its dependence on \gamma and \epsilon is also worse than previous bounds”
>
> A: The form of the bound in Theorem 1 does depend on \ln(1/\delta) as well (see the full proof). But, as we already mentioned in the paper, the form of the bound that we used suppresses logarithmic dependence. The dependence on \gamma and \epsilon is close to Strehl et al. (2006). As for BOSS (Asmuth 2009), the form of the bound in Theorem 3.1 of the original paper---which seems to be better---is not actually better because its final form also depends on $B$, the Bayesian concentration sample complexity that also depends on \gamma and \epsilon. The final form of BOSS’s bound is in Section 3.2 of the original paper, which additionally depends on another constant $c$ and does not show better dependence on \gamma and \epsilon than our bound. Note that although lifelong learning has been widely studied in supervised learning domains and performance bounds are available, our paper is the first work in lifelong reinforcement learning that derives an explicit performance bound, showing how a learned prior can affect the sample complexity of model-based RL algorithms.
>
> Q5: “Also, their use of model-predictive control is common in current ML applications, but, again, this is not mentioned or discussed (e.g., the length of the future horizon and how it is selected).”
>
> A:  Thank you for making this observation; we now include in the paper (Section A.4.3 in the appendix).
>
> Q5: “While the method compares well to simple baselines it is hard to assess performance relative to more recent work such as Liu et al., "Taming MAML: Efficient Unbiased Meta-Reinforcement Learning" and the other baselines measured in it.”
>
> A: First, we would like to point out that, as far as we are aware, LPG-FTW (Mendez et al.) is still the state-of-the-art lifelong RL algorithm. We also modified a HiP-MDP baseline to our model-based setting for a fair comparison. And EWC is one of the state-of-the-art single-model-type lifelong RL algorithms as pointed out in Mendez et al.’s paper. We therefore disagree with the reviewer that these are “simple” baselines.
>
> Secondly, Liu’s paper and the other baselines in that paper are targeted at meta-learning settings. Meta-learning typically focuses on scenarios where a large batch of tasks is available for training and evaluation is done either on the same batch or on a target task. Our method, however, focuses on problems where the tasks arrive sequentially instead of having a large number of tasks available at the beginning of training. This sequential setting makes it hard to accurately infer the hidden parameters, but opens the door for algorithms that support backward transfer. That is, lifelong learning is not quite the same setting as transfer (which is what meta-learning targets). We discussed this in the related work section.
>
> Q6:  “However, I do not find that the level of innovation in this combination of approaches suffices for publication at ICLR, nor did I find the theoretical or experimental results of sufficient interest (see comments above).”
>
> A: Here we refer the reviewer to the first comment replying to all reviewers about the novelty and contributions of the paper. We simply reiterate that our novel algorithm outperforms the previous state of the art, LPG-FTW, which was published at NeurIPS 2020, and the other previous state of the art, which was published at NeurIPS 2017, etc. We also believe the problem setting to be important as one of the major questions of lifelong reinforcement learning is to decrease the amount of experience needed to learn new tasks, but very little work has been done about exploration in lifelong reinforcement learning in recent years.

---

> > ### Comment · Reviewer_7Ubo · 2021-11-26
> > **The new version does not allevaite my concerns, in fact it exacerbates them**
> >
> > In Theorem 1 (and the new Theorem 2 in the appendix) the authors present a bound on sample complexity whose dependence on \delta is 1/\delta and for which deviation from the optimal value function also depends on 1/\delta. However, similar results, as summarized in Strehl et al. “Reinforcement Learning in Finite MDPs: PAC Analysis” JMLR 2009, depend on \delta through log(1/\delta), which is exponentially better in terms of \delta. The revised proof of Theorem 1 in the appendix did not shed further light on this issue, and, in fact, is not a full proof. Since this is essentially a bound on the single instance case, it should be comparable at least to previous results, and it seems not to be. Furthermore, the dependence of the theorem on S is S^2, while recent results depend on S rather than. In fact, the best recent results I am aware of are O(SA\log(1/\delta)/(1-\gamma)^{5.5}\epsilon^2) (Zhang et al Model-Free Reinforcement Learning: from Clipped Pseudo-Regret to Sample Complexity, NeurIPS 2021), which is significantly better that the rates provided in the present paper.
> >
> > Additionally, the paper does not present a true meta-learning bound, but rather a single instance bound based on an assumed posterior consistency. This does not deal with the most important aspect of meta-learning, namely how the posterior improves in time and enables successful transfer. The authors state at the end that their work is one of the first to deal with performance bounds for lifelong RL. However, their results are far weaker, conceptually and technically,  than those available for supervised learning, and, as I mentioned, do not constitute real lifelong learning bounds as stated above.

---

> > > ### Author Response · Authors · 2021-11-28
> > > **Thank you for your response**
> > >
> > > First, we would like to point out that BLRL is built upon BOSS (Asmuth 2009) and so does the sample complexity analysis. The bound on sample complexity proposed in BOSS does depend on 1/\delta as well as log(1/\delta). We hope the reviewer can refer to the exact name of the theorem in paper “Reinforcement Learning in Finite MDPs: PAC Analysis” because RMAX apparently depends on S^2, which is the algorithm that BOSS is built upon. We believe the right bound that should be compared with is the one proposed in BOSS, as we use the exact same algorithm procedures except that we use the learned world model to initialize the task-specific prior. The comparison with the best recent single-task model-free bound is not fair for our method. Therefore, We respectfully disagree with the reviewer that the bound is far weaker. Moreover, the reviewer wants us to provide a meta-learning bound. But as we already mentioned in our first response, meta-learning is a different setting from lifelong learning. Here we care more about the agent's performance on each new task that arrives sequentially and how the accuracy of the previous learned common knowledge would affect the learning efficiency for each new task, which is shown by our theoretical results.

---

> > > > ### Comment · Reviewer_7Ubo · 2021-11-29
> > > > **I still do not see the theoretical advantage of lifelong learning**
> > > >
> > > > Thank you for your explanation. I still find it hard to understand the main focus of the paper. First, your framework is Bayesian and yet you present standard PAC-MDP bounds that were developed in a non-Bayesian framework. While it is indeed possible to derive such bounds for Bayesian or pseudo-Bayesian algorithms (such as Thompson sampling), this is not clear from your description and should be clarified. As you say, your bounds apply to a single task setting, in the context of lifelong learning. However, in the latter context I would like to see that the posterior improves in time, otherwise what is the purpose of lifelong learning. Simply assuming that it is consistent does not address this crucial issue. As I mentioned, in the single task setting there are very tight recent bounds, specifically Theorem 2 in the paper I mentioned Zhang et al., "Model-Free Reinforcement Learning: from Clipped Pseudo-Regret to Sample Complexity" (NeurIPS 2021, arXiv:2006.03864v3) provides the tightest bounds I am aware of. While it is fine and good to compare to BOSS there have been many advances in bounds since then and it is not clear why your approach should be preferred. Moreover, you state in your response that "We believe the right bound that should be compared with is the one proposed in BOSS, as we use the exact same algorithm procedures except that we use the learned world model to initialize the task-specific prior." However, I did not see how the learned prior enters your sample complexity analysis, which is the essential feature. I may be missing something, but this crucial point should be stated clearly and convincingly (it does not suffice to say that if the prior is closer to the true prior then performance is improved). Overall, the basic ideas of the paper are interesting, but I believe that more work needs to be done in order to demonstrate their theoretical advantages, which, as I understand, is one of the main motivations of the work. While the authors argue that they do not deal with meta-learning, still, I do not see the theoretical benefit of lifelong learning from their analysis, even if the simulations do so empirically. Also, as stated by other reviewers the writing should be improved, and full assumptions, definitions and proofs need to be provided.

---

### Official Review · Reviewer_FNYu · 2021-11-02

**Correctness:** 4
**Technical Novelty And Significance:** 2
**Empirical Novelty And Significance:** 2
**Recommendation:** 3
**Confidence:** 4

**Main Review:**

############## Strengths ##############

1. The overall idea of model-based lifelong RL is very relevant, particularly since lifelong RL precisely seeks to reduce sample complexity
2. The high-level idea of replacing the task model with the world model whenever the task model is uncertain is intuitively appealing
3. The use of the task model permits deriving reduced sample complexity bounds thanks to the Bayesian formulation

############## Weaknesses ##############

1. The low performance of the agent in the more complex MuJoCo evaluations makes the results unconvincing
2. The technical approach is very closely tied to existing works, in particular HiP-MDPs and variants and BOSS
3. There are no comparisons to existing model-based lifelong RL methods

############## Arguments ##############

The primary contribution of this work is the introduction of one of the first approaches for model-based lifelong RL. Since one of the key desiderata of lifelong RL is to decrease the amount of experience needed to learn new tasks (forward transfer), using model-based techniques, which are inherently more data efficient than model-free techniques, seems to be a promising direction.

However, the proposed method itself is fairly incremental. In particular, it heavily hinges on BOSS (as an exploration technique) and HiP-MDPs (as a multi-task model-based model). While the use of these models permits relatively simple adaptations of the proofs of BOSS to demonstrate decreased sample complexity (which is a nice plus), it does make the work contain very little technical insights. In particular, when adapting a multi-task method (HiP-MDPs) into a lifelong method, I would have expected to see considerable effort in designing a technique for updating the shared model with knowledge from new tasks. However, the authors simply train this shared model in a multi-task fashion with data from all tasks. This is not only costly in terms of memory footprint (requiring to store vast amounts of data for each task), but is also computationally inefficient, since the multi-task training step is executed iteratively _for each task_ (O(n^2) cost). The one piece of technical insight offered in the submission that I found interesting was the idea of replacing the task model with the world model whenever the task model is not yet well trained. Unfortunately, the approach itself is very heuristic (relying on thresholds on the uncertainty across sampled models) and the heuristic choices are not discussed in detail or validated empirically (e.g., via ablative tests or sensitivity analysis on the thresholds). I intuit that the method is quite sensitive to these thresholds: if they are too high, the task model will almost always be used, even if it's uncertain; if they are too low, the world model will be used too frequently and task performance will likely degrade by reverting to the "average" world model. Would it not be possible to do some sort of soft combination (of the task and world model) weighted by the uncertainty, instead of a hard selection of one or the other?

However, my main concerns with the submission lie on the empirical evaluation. The first major concern is that all the agents evaluated on MuJoCo domains achieve very low performance, as compared to the results in the papers cited in the submission for the experimental design of these tasks (Mendez et al. and Wang et al.). While it is clear that the reduced performance stems from the choice of using fewer environment interactions (which itself is a nice choice, given the use of model-based techniques), this does raise questions about the conclusions drawn from these results. In particular, how useful or informative are forward and backward transfer if the agent hasn't really learned any meaningful behaviors? It would be useful to include videos of the learned behaviors to assess whether the transfer results are in any way significant from a behavioral perspective, or if they're simply minor reward increases across poor behaviors.

My second major concern is the choice of baselines for these evaluations. On one hand, the authors chose to only compare against model-free lifelong RL techniques. In this setting, it should certainly be expected that the model-based approach outperforms those baselines simply by the nature of the underlying techniques. This is not novel insight. Yet the authors claim these as general improvements over existing lifelong RL methods, which seems like a stretch. Instead, the authors should have considered existing model-based lifelong RL methods (e.g., [1]) for a more apples-to-apples comparison. Note that [1] is a task-agnostic method, so this would require some adaptation to handle the setting where the agent is given access to task indicators. On the other hand, the authors make claims about forward transfer, but in the MuJoCo domains there is no comparison to a single-task or no-transfer baseline, like used in the box-jumping task. Such a baseline is critical for assessing whether the approach is actually achieving transfer across tasks, since improvements w.r.t. single-task training are precisely what demonstrate transfer. As one additional comment regarding the evaluations, there is no information about the implementation details of any of the baselines, including their model architectures and hyper-parameters. How were these chosen to guarantee a fair comparison?


############## Additional feedback ##############

The following points are provided as feedback to hopefully help better shape the submitted manuscript, but did not impact my recommendation in a major way.

Intro
- I wonder if the example of different houses and toothbrushes matches the HiP-MDP formulation introduced immediately after
- The intro is fairly clear and describes the solution approach well.
- I'd suggest including an example that more closely matches the HiP-MDP formulation, since this is the formulation adopted throughout the paper.

Sec 4
- The ideas seem to be very closely related to the original HiP-MDP papers, especially the BNN extension of Killian et al. (2017)
- The notation for the BNN needs a fair bit of work. The authors never explain what the "particles" are. Are these the (s,a,r,s') tuples sampled from the sequences given by the combination of CEM and the BNN? This becomes increasingly relevant in 4.1 where the authors define their approach to backward transfer. My understanding is that "aleatory variability" is modeled as the BNN's internal variance, whereas the epistemic uncertainty is measured as the variance in the (mu, sigma) output by the BNN across sampled particles. Is this understanding correct?

Sec 5
- The gird-world evaluation shows nothing about forgetting/backward transfer.
- Box-jumping: Why no comparison to a single-task variant of the solution? This is required to assess forward transfer. Also no backward transfer measure.
- MuJoCo: again, no single-task learner, so it's unclear if there's forward transfer. Plus, the rewards are very low, so it seems that even VBLRL is not solving the tasks. How useful are these results then? Even though VBLRL is the best, it's not really achieving meaningful behaviors.
- I disagree with the claim that the model "cannot" suffer from forgetting, since certainly the wrong choice of threshold for backward transfer could lead to forgetting.
- How was this hyper-parameter chosen?

Typos
- Sec 2, second paragraph: the task facing a single agent -> the agent facing a single task?

[1] Nagabandi et al. Deep online learning via meta-learning: Continual adaptation for model-based RL. ICLR 2019.


**Summary Of The Paper:**

This submission presents an approach for Bayesian model-based exploration in a lifelong RL setting, building upon existing approaches for Bayesian exploration (BOSS) and Bayesian multi-task modeling (HiP-MDPs). The approach keeps separate models for sampling transitions and rewards for each task, and each task model is drawn from a shared prior that models the distribution over tasks. The method continually updates the shared model with data from all observed tasks and the task-specific model with data from the current task. To achieve backward transfer, the approach replaces the task model with the shared model whenever the task model has not been sufficiently trained on a particular state-action pair.


**Summary Of The Review:**


Unfortunately, I recommend the rejection of this work. While I agree with the premise of the submission that model-based lifelong RL is a relevant area of research, with potential implications on real-world applications of lifelong RL, the submission as it stands appears to not be ready for publication. On the technical side, the approach seems to add just a few incremental changes to multi-task HiP-MDPs to adapt them to the lifelong setting. This on its own is perhaps relatively minor, since the novelty comes from adapting it to a new problem setting. However, such technically incremental contributions should generally be accompanied by strong empirical evaluations, which is not the case in this work. In particular, the low overall performance of all agents on MuJoCo domains suggests that none of the agents are learning to achieve meaningful behaviors, which raises questions about the conclusions reached by the authors. Moreover, the authors should have compared (at least qualitatively, but ideally also empirically) to existing work in lifelong model-based RL. On the flip side, the submission does include an interesting insight of replacing the task-specific model with the shared model whenever the task model is uncertain.

---

> ### Author Response · Authors · 2021-11-19
> **Response to Reviewer FNYu (Part 1)**
>
> Q1: “The low performance of the agent in the more complex MuJoCo evaluations makes the results unconvincing” “how useful or informative are forward and backward transfer if the agent hasn't really learned any meaningful behaviors?”
>
> A:
> 1. Our numbers should not be directly compared to those reported in prior work, for two reasons. 1) We use shorter episodes for each task. 2) We substantially reduced the number of environment interactions (episode length 1000 -> 100/200, iterations 200 -> 100). The resulting cumulative reward would therefore decrease a lot because it’s dependent on the number of steps within an episode (LPGFTW uses 1000, we use 100). The cumulative reward results for single-task RL methods shown in Wang et al.’s paper use 5000 steps for each episode, while we use 200 steps. Note that if you divide the results in those papers by the corresponding translation factor, you get values similar to our results (that is, divide by 10 for halfcheetah in LPG-FTW, divide by 25 for walker/hopper in Wang’s paper). The reason we used different settings in our work was to increase the difficulty of lifelong learning and evaluate the behavior of these algorithms in the low-data regime. As detailed in the appendix, we used different reward settings for walker and hopper to make them amenable to model-based RL following Wang et al. (2019), compared to Mendez et al. 's work. In brief: The performance we report is actually quite close to that in prior work.
>
> 2. In many single-task model-based RL papers such as PETS, the algorithms are evaluated in a setting where the true underlying reward function is known, so the agent only needs to predict the dynamics (next state) and do planning. In such settings, we agree it’s essentially unfair to compare these methods to model-free RL algorithms, which do not benefit from knowing reward functions. However, in our lifelong problem setting, we make the reward functions unknown to the agent, which we think is a more general and reasonable setting when different tasks have different reward functions. And we train our model to also predict the reward as well as the next state, greatly increasing the level of difficulty for model-based planning as they require much more accurate predictions. We think this setting makes the comparison with model-free methods fair enough and it may lead to the performance drop compared with some other model-based RL papers that assume the reward function is known.
>
> 3. In our updated appendix, we attached a link to the video https://youtu.be/I7RAT6g9v5w that show the agent trained by our algorithm learns some meaningful behaviors at the current setting within limited steps, and performs better compared with the behaviors learned by other baseline algorithms. With our current settings, our method adapts faster on new tasks thanks to forward transfer. Using a fixed and limited number of environment interactions illustrates the superior performance of our algorithm compared to baselines. We believe this observation should resolve the reviewer’s concerns.
>
> Q2: “the authors chose to only compare against model-free lifelong RL techniques.” “the authors should have considered existing model-based lifelong RL methods (e.g., [1])”
>
> A:
>
> 1. The HiP-MDP baseline we compared against is, in fact, a model-based lifelong RL method. (Killian et al., also uses BNNs to construct the dynamics model.) As we mentioned in Section 5.2, we replaced the DDQN algorithm used in Killian et al.’s paper with CEM planning based on BNNs for a fair comparison, which is a standard model-based RL technique and we used the exact same parameters for CEM and BNN as in VBLRL.
>
> 2. As we mentioned in the reply to the previous question, we think it is a fair comparison with model-free lifelong RL methods as we take the reward functions as unknown unlike previous comparisons to model-based algorithms (PETS, as well as Nagabandi’s paper mentioned by the reviewer). Our framework is able to be combined with model-based methods and thus greatly increase sample efficiency for lifelong RL, which is one of the main advantages and a principal motivation for our approach.
>
> 3. As far as we are aware, our work is the first paper to look at model-based lifelong deep RL in the HiP-MDP setting. The model-based continual RL paper the reviewer referred to (a) focuses on discovering when new tasks were encountered in the absence of task indicators, which is also pointed out in LPG-FTW paper. In contrast, we consider the same setting as was studied in the LPG-FTW paper, which is to increase forward/backward transfer performance when the task indicators are given and we know when new tasks arrive. We add and clarify this point in our updated related work section. (b) The authors of [1] did not release their source code and we were not able to reproduce it.

---

> ### Author Response · Authors · 2021-11-19
> **Response to Reviewer FNYu (Part 2)**
>
> Q3: “the authors make claims about forward transfer, but in the MuJoCo domains there is no comparison to a single-task or no-transfer baseline, like used in the box-jumping task.” “Such a baseline is critical for assessing whether the approach is actually achieving transfer across tasks, since improvements w.r.t. single-task training are precisely what demonstrates transfer.”
>
> A: Thank you for raising this point. Single-task baselines are straightforward and we included the single-task baselines in the updated version of our paper. Compared to single-task baselines, our method exhibits superior performance in all scenarios and demonstrates positive forward transfer. The reason we did not include these results originally is that we made use of existing comparison procedures. Except for the number of interactions with the environment, we followed the same experimental protocols as in the LPG-FTW paper, which is the state-of-the-art lifelong RL algorithm as far as we are aware. With the same goal of evaluating forward transfer, they did not compare to a single-task baseline and instead assessed forward transfer by observing the faster learning speed compared to other baselines.
>
> Q4: information details of the baselines
>
> We have added more details in Section A.4.3 in the appendix.
>
> Q6: “it heavily hinges on BOSS (as an exploration technique) and HiP-MDPs (as a multi-task model-based model). While the use of these models permits relatively simple adaptations of the proofs of BOSS to demonstrate decreased sample complexity (which is a nice plus), it does make the work contain very little technical insights.” “the approach seems to add just a few incremental changes to multi-task HiP-MDPs to adapt them to the lifelong setting. This on its own is perhaps relatively minor, since the novelty comes from adapting it to a new problem setting.”
>
> A: We listed the contributions and novelty of our paper in the section above for all reviewers. We hope this clarification alleviates your concerns. Note the updated algorithm section (a new figure 2) as well as the full proofs included the Appendix A.5 & A.6 in the updated version of our paper.
>
> Q7：Replacing the task model with the world model whenever the task model is not yet well trained. The heuristic choices of the threshold are not discussed in detail or validated empirically.
>
> A: We agree with the reviewer’s point that it’s hard to find such thresholds for each task in practice. However, as we already mentioned in the appendix, during planning, for each prediction, we instead choose to compare the uncertainty of the output mean and variance of the world model and the task-specific model, and then choose the one with lower values, which indicates higher confidence. In this way, we do not need to manually choose the value of threshold for each task. And this simple approach turns out to perform well as shown in the results of gym Mujoco tasks.
>
> Q8: Notation for BNN and backward transfer part.
>
> A: Thank you for pointing out this shortcoming. We have added that in the corresponding sections of the updated version of our paper.

---

> > ### Comment · Reviewer_FNYu · 2021-11-21
> > **Response to author rebuttal (1/2)**
> >
> > I thank the authors for their detailed responses. Unfortunately, only two of my concerns seem to have been addressed satisfactorily: the addition of the STL baselines and the explanation of the fact that there is no need to manually pick thresholds for achieving backward transfer. However, a number of significant concerns remained unaddressed, and some new concerns were raised by the authors responses. In particular, the authors failed to address my comment regarding the computational efficiency of the proposed approach, and more critically made several factually incorrect claims about the existing literature. With this in mind, I will maintain my original score as I do not believe that the submission in its current form is ready for publication. Please find below a detailed response to the authors' rebuttal.
> >
> > - **Regarding general comments**. I would caution the authors about making bold statements such as "our paper is the first work in lifelong reinforcement learning that derives an explicit performance bound". This is factually incorrect: at least "PAC-inspired Option Discovery in Lifelong Reinforcement Learning" by Brunskill and Li (2014) obtained performance bound in a lifelong RL setting.
> > - **Regarding response to reviewer eZuT**. The authors again make a bold statement that "no existing work successfully applies a lifelong RL algorithm on image input domains". This again seems to be factually incorrect: [1,2,3] are some examples of lifelong deep RL methods that were indeed applied to image domains.
> >
> > 1. **MuJoCo evaluation**. I appreciate the clarification regarding the scale of the returns in MuJoCo domains. This does raise one additional concern. While this cannot be verified due to the lack of source code provided along with the submission, I would imagine that the authors are using the same time-step length as in the original OpenAI Gym environments. In particular, this value is set to dt=0.002 for Hopper and Walker-2D domains, which means that the authors are evaluating methods for their abilities to hop/walk _for a total of 0.4s_. This incredibly short horizon seems to still undermine the results, as suggested in my original review. This seems to be validated by the new video results, which show that the hopper simply learns to fling itself forward and to the ground (which is a policy that would be useless if a longer horizon were considered). My recommendation to the authors would be to evaluate on trajectories with the original horizons, which were designed to evaluate the ability of the methods to learn policies that worked in the relatively long term, while limiting the _total_ number of interactions with the environment during training.
> > 2. **Choice of baselines**. While I agree that comparing against model-free baselines is _fair_, my comment about baselines was that this comparison yields no new _insights_: of course model-based (lifelong) RL methods should be more sample efficient than model-free (lifelong) RL methods, just like their single-task counterparts. However, I should note that HiP-MDPs are _not_ a model-based lifelong baseline, but a batch multi-task baseline that the authors adapted to the lifelong setting in the most naive way (which serves as perhaps a nice ablation, but not a proper baseline). Indeed, Nagabandi et al. consider the task-agnostic setting (as I alluded to in my review); I appreciate that the authors were able to include the citation in the revised draft, at least. I would encourage them to dig deeper into this connection instead of reciting the comparison from Mendez et al., given that the work of Nagabandi et al. is (to my knowledge) the only existing lifelong model-based RL method and seems like a very relevant related work.

---

> > > ### Comment · Reviewer_FNYu · 2021-11-21
> > > **Response to author rebuttal (2/2)**
> > >
> > > 3. **Single-task comparison**. I think the addition of the single-task results is quite significant and does highlight the fact that the method (and indeed most of the baselines) are achieving forward transfer w.r.t. STL. I should note that the LPG-FTW paper _does_ indeed compare against STL in every one of their experiments.
> > > 4. **Details about baselines**. For LPG-FTW and EWC it is unclear how the search over the suggested hyper-parameters was conducted (search over all tasks, search over a subset of the tasks...?). **I am very concerned about the statement that LPG-FTW and EWC use the same architecture as in the original paper**. Those architectures were simple linear models, while VBLRL is using far more complex BNNs. Could the authors discuss any other steps that might make this comparison more fair? This is particularly important as these are the only true lifelong baselines the paper compares against.
> > > 5. **Novelty**. As far as I can tell, the general response does not point to any additional novelty than I alluded to in my original review. While I still contend that this on its own wouldn't be a cause for rejection, it does appear to be a limitation of the submission: there is not much technical novelty or insight.
> > > 6. **Backward transfer thresholds**. This does make the approach seem much more robust. Given the importance of this, I would encourage the authors to at least briefly describe this choice in the main paper.
> > >
> > >
> > > [1] Kirkpatrick et al. Overcoming catastrophic forgetting in neural networks. PNAS, 2017.
> > >
> > > [2] Schwarz et al. Progress & compress: A scalable framework for continual learning. ICML, 2018.
> > >
> > > [3] Rolnick et al. Experience replay for continual learning. NeurIPS, 2019.

---

> > > > ### Author Response · Authors · 2021-11-22
> > > > **Response to post rebuttal comments from Reviewer FNYu**
> > > >
> > > >
> > > > We appreciate the reviewer’s quick and thoughtful response!
> > > >
> > > > Q1: Mujoco evaluation.
> > > >
> > > > We attached our source code here \url{https://drive.google.com/file/d/1nq55JWVysZCPNOSILsAyiFnfWM1KP0zb/view?usp=sharing}. The time-step length we used for Halfcheetah is 0.05, while for both Hopper and Walker-2D are 0.008 (instead of 0.002), which gives a total of 1.6s  for hopper/walker in the experiments. We believe 200 steps horizon is a reasonable setting also because similar horizon settings (half cheetah/walker) are also applied in some recent deep RL papers that require a limited total number of samples [1][2]. The policy learned by our agent shown in the video is not perfect but still meaningful under the current short-horizon setting, considering the total number of interactions is significantly decreased compared to those single-task RL experiments. We did not experiment with the original task horizon as we limit the total number of interactions for each step to only 20000 (hopper/walker). If we set 5000 steps for each task, the agent would interact with the environment for only 4 episodes (reset), which we do not think is a good setting for learning.
> > > >
> > > >
> > > > Q2: About the statements made in the comment.
> > > >
> > > > We apologize for the confusion here. We have revised our comment. As far as we are aware, our paper is the first work in lifelong reinforcement learning under the same HiP-MDP setting that derives an explicit performance bound.
> > > >
> > > > Q3: Choice of baselines
> > > > First, please note that multitask HiP-MDPs are a problem formalization, not an algorithm. Besides, we believe the HiP-MDP baseline (Killian et al.) can be viewed as a model-based lifelong RL algorithm: 1. They use BNN to train a dynamics model and plan based on that (model-based) 2. Under the same HiP-MDP setting,from multi-task/meta learning to lifelong learning is a natural extension that the agent uses the same method to continually adapt to new tasks.
> > > >
> > > > Q4: Details about baselines
> > > > Here, we simply try different hyper-parameters suggested by the original paper (which is not a large number), and choose the one with the highest returns for each domain. We use the same architecture as in the original paper (the released source code) for LPG-FTW and EWC. Note that NPG and Episodic Reinforce are the only two methods provided that are able to be used as the base learner of LPG-FTW. We did not find a way to combine their methods with more complicated deep rl algorithms and that’s beyond this paper’s scope.
> > > >
> > > > Q5: Computational efficiency
> > > > 1. When evaluating forward transfer on new tasks, we initialize the task-specific model with parameters of the world model and train using data only from the new task. When evaluating backward transfer on previously visited tasks, we do not do further training but instead propose a novel way to combine the task-specific model and world model and make decisions.
> > > > 2. The lifelong RL setting we consider in this paper is HiP-MDP, where the hidden parameters \omega are sampled i.i.d. instead of controlled by unobserved dynamics like DP-MDP[3]. Therefore, the most reasonable way for training on a batch of previously visited tasks in this setting is to train them in a multi-task fashion (i.e. treat them equally). Another way of viewing this is that, each time after finishing visiting a number of tasks, the problem of training a world-model that captures the joint distribution of these tasks and can achieve faster adaptation on new tasks is a multi-task/meta learning problem. Our approach of updating the world-model is therefore straightforward and reasonable under this setting, which is also empirically effective. We agree with the reviewer that a more computational efficient way of doing this may exist, and we leave this to future work.
> > > >
> > > > Q6: Backward transfer thresholds
> > > > Thank you for the suggestion! We have added the description to the main paper.
> > > >
> > > > [1]. Efficient off-policy meta-reinforcement learning via probabilistic context variables. ICML, 2019
> > > > [2]. MetaCURE: Meta Reinforcement Learning with Empowerment-Driven Exploration. ICML, 2021
> > > > [3]. Deep Reinforcement Learning amidst Lifelong Non-Stationarity. ICML 2021

---

> > > > > ### Comment · Reviewer_FNYu · 2021-11-23
> > > > > **Additional responses**
> > > > >
> > > > >
> > > > > 1. While I continue to agree with the choice of using fewer interactions with the environment to test the sample efficiency of the proposed method, I am still unconvinced by the authors' claims. First, in the source code provided by the authors, the timesteps are _not_ set to the values mentioned in the author response, but rather to the original values of 0.01 (for halfcheetah) and 0.002 (for walker and hopper). Second, the hopper video uploaded by the author seems to confirm that the length is substantially less than the reported 1.6 sec (see 2 sec video demonstrations of Mendez et al. [here](https://github.com/Lifelong-ML/LPG-FTW/blob/master/videos/mujoco/hopper_bodyparts.mp4) for comparison). Third, there are no walker videos included, so there is no way to validate that indeed the learned behaviors are meaningful. The authors claim that using a horizon of 5,000 would be too much to permit learning with only 20,000 samples. While this very well may be true, there are surely multiple choices between 200 and 5,000 that would permit 1) learning meaningful behaviors while 2) evaluating the sample efficiency of the methods in question.
> > > > > 4. I insist that using the original linear models for LPG-FTW and EWC does not lead to a fair comparison. Since these are the primary baselines that the method is compared against, I would encourage the authors to expend more efforts in comparing the methods more fairly. It is possible that LPG-FTW/EWC are only underperforming due to the use of linear models, which has nothing to do with the claims in this paper. **Demonstrating that the proposed method has advantages w.r.t. existing methods in a fair evaluation setting should surely not be out of scope.**
> > > > > 5. Training in a MTL fashion is not necessarily "the most reasonable way", though certainly it is the simplest way. Since there aren't many technical novelties in this work, the authors might consider developing a mechanism that reduces the computational cost of repeatedly training on batches across all previously seen task, which is _extremely_ expensive in a lifelong setting.

---

> > > > > > ### Author Response · Authors · 2021-11-23
> > > > > > **Response to additional responses from reviewer FNYu**
> > > > > >
> > > > > > Thanks for responding. Please note that in the source code, details about the environments are mainly in $\textbf{gym-extension-mod/}$ folder, where we do use 0.05 time-step length for halfcheetah, and 0.008 for Hopper and Walker2D.

---

### Official Review · Reviewer_eZuT · 2021-11-05

**Correctness:** 1
**Technical Novelty And Significance:** 2
**Empirical Novelty And Significance:** 2
**Recommendation:** 5
**Confidence:** 2

**Main Review:**


- Clarity of the paper should be improved significantly. The current version makes understanding hard.
     - The paper does not describe the notion of many notations (e.g., the indices j and k in Figure 1.) and denotes some distributions in its short-form, e.g., q_e or q_\theta^{m_i}, without introducing its full form. These are not just a few, but observed broadly across the entire paper, perhaps Section 3 requires the most significant improvement.
     - The paper also explains the proposed method by directly explaining lines of the pseudo-code without discussing the big picture or the rationale behind the design.
     - It also skipped describing the full joint distribution of the model which is essential in describing such probabilistic models.
- The key idea of the proposed method is quite simple. It's like applying Variational Continual Learning to the world model learning in the RL setting. It can also be understood as Bayesian meta-learning but with sequential task exposure.
- The experiment is also quite simple. The first two toy tasks are quite too toyish, very low-dimension and tow task complexity. The last MuJoCo experiments show the superiority of the proposed model.
- The algorithms (e.g., variational continual learning, Bayesian meta learning, and mean-field variational inference) on which the proposed method is based are known to be not scalable in more complex settings like high-dimensional data (like image), many sequential tasks (e.g., 1000 or more tasks), and the mean-field approximation has significant limitation in its expressiveness. So, I'm somewhat doubtful if this method can be an important milestone toward more realistic and complex settings.

**Summary Of The Paper:**

The authors proposes a Hierarchical Bayesian approach for lifelong RL. The global world-model posterior models the world model shared across tasks and the task-specific model learns the dynamics within a specific task. The task-specific model achieves forward transfer by initializing from the global world model. The authors use mean-field variational approximation to scale the proposed model. Also, the authors introduce sample complexity analysis. The method is evaluated on two toy tasks (grid-world and box jumping) and one on MuJoCo simulator and showed superior performance to the previous works.

**Summary Of The Review:**

The writing clarity requires a significant improvement. It's currently a major drawback hindering the understanding of the proposed model (I understood at the later part of the paper but it was hard until reaching there). The evaluation is quite simple and uses toy tasks. I'm doubtful about the potential of the proposed model to extend to more realistic and complex settings such as image inputs.

---

> ### Author Response · Authors · 2021-11-19
> **Response to Reviewer eZuT**
>
> Q1: “Clarity of the paper should be improved significantly. The current version makes understanding hard. The paper also explains the proposed method by directly explaining lines of the pseudo-code without discussing the big picture or the rationale behind the design etc.”
>
> A: Thank you for pointing this out. We revised the paper, and we hope the new version is clearer. If not, please let us know; we are happy to revise until you are satisfied as to clarity. In addition, we discussed the big picture behind the method in our introduction, the first two paragraphs of section 3, as well as the new Figure 2 and the explanation together with it. Please see our reply to all reviewers for more information.
>
> Q2: “The key idea of the proposed method is quite simple. It's like applying Variational Continual Learning to the world model learning in the RL setting. It can also be understood as Bayesian meta-learning but with sequential task exposure.”
>
> A: Thank you for appreciating our method's simplicity, we worked hard to make it clear and easy to understand. Please see our comment replying to all reviewers about the novelty and contributions of the paper.
>
> Q3: “The first two toy tasks are quite too toyish, very low-dimension and low task complexity.”
>
> A: The first gridworld task is a finite MDP setting and we aim to evaluate the first proposed algorithm BLRL here, compared with the single task baseline BOSS. We used this and the jumping task (VBLRL) for illustrative purposes, relying on the 6 different Mujoco domains for a more thorough evaluation, but can remove them if the reviewer would like.
>
> Q4: “I'm doubtful about the potential of the proposed model to extend to more realistic and complex settings such as image inputs.” “The algorithms (e.g., variational continual learning, Bayesian meta learning, and mean-field variational inference) on which the proposed method is based are known to be not scalable in more complex settings like high-dimensional data (like image), many sequential tasks (e.g., 1000 or more tasks),”
>
> A: Firstly, we would like to point out that we compare to the standard benchmark lifelong RL settings introduced in Mendez et al. (2020), where our algorithm outperforms the state-of-the-art lifelong RL algorithm LPG-FTW as well as the most recent HiP-MDP method with model-based modifications. Secondly, very few existing work successfully applies a lifelong RL algorithm on image input domains such as the DeepMind control suite or Atari games. In particular, LPG-FTW is also not evaluated on image-based RL environments. However, we want to point out that VBLRL does have the potential to extend to image input settings. As we mentioned in the appendix section A.7.1, the CEM planning module can be replaced with any other model-based planning technique that works well in image environments, such as Dreamer (Hafner et al. 2020). We leave this extension to future work. In any case, we disagree with the reviewer that image-based settings are always more realistic. Mujoco is a physics simulator; there are many real domains where something like a joint space, rather than an image, is the natural state representation.

---

### Author Response · Authors · 2021-11-19
**Main contributions/novelty (reply to all reviewers)**

Here we list the main contributions/novelty of our paper:

1. Our paper focuses on the exploration problem in lifelong reinforcement learning, which is critical but has not been a research focus in recent years. Compared to single-task reinforcement learning, sample-efficiency is even more important in lifelong reinforcement learning (decrease the amount of experience needed to learn new tasks, as pointed out by reviewer 2), which makes the exploration problem more crucial and valuable to study. Our algorithm shows its superiority to the state-of-the-art alternatives in several continuous control benchmark lifelong RL tasks.

2. Although lifelong learning has been widely studied in supervised learning domains and performance bounds are available, our paper is the first work in lifelong reinforcement learning (with HiP-MDP assumption) that derives an explicit performance bound, showing how a learned prior can affect the sample complexity of model-based RL algorithms. This bound can be extended to any other Bayesian transfer setting in model-based RL (lifelong learning, meta learning, etc.). We include a full proof in the appendix of the updated version of our paper (A.5 & A.6).

3. The proposed two model-based lifelong RL algorithms (BLRL for finite MDPs & VBLRL for more general problems) provide a novel way to separately estimate different kinds of uncertainties in the HiP-MDP setting. We add a detailed explanation of this fact in Figure 2 in the updated version of our paper.

4. Compared to PETS (Chua et al., 2018) and other previous CEM-based RL methods, we made a significant modification that is more amenable to the lifelong RL setting with a hierarchical Bayesian posterior and CEM planning: Instead of maintaining a large number of neural nets (>= 30) for each task, which is unrealistic in lifelong learning, we use one task-specific BNN for each task that can be seen as an infinite neural network ensemble. In this way, we only have to train one task-specific neural network for each task and we can sample an unlimited number of different action sequences to cover more possibilities as needed. In contrast, in PETS, samples are limited by the number of neural networks in the ensemble. We explained this in the last paragraph of section 4.0, page 5&6.

5. We propose a novel idea that the world-model posterior trained on more tasks can help infer the “unknown” part of the task-specific model for previous visited tasks, thus is also beneficial for backward transfer. Our experimental results on Mujoco benchmark tasks show that VBLRL can achieve positive backward transfer, which has not been demonstrated in previous lifelong deep RL algorithms.

6. Our first experiment on gridworld shows that BLRL achieves positive forward transfer in the finite MDP setting. In the following jumping task and standard Gym Mujoco benchmarks used by previous lifelong RL papers, our algorithm outperforms state-of-the-art lifelong RL methods (LPG-FTW and EWC), a similar Model-based lifelong RL baseline (modified HiP-MDP), as well as single-task model-based baseline within limited data for each task.

---

### Decision · Program_Chairs · 2022-01-20

**Decision:**

Reject

**Comment:**

The topic of this paper is timely and important.  However, ultimately the reviewers remained unconvinced that this paper provides a sufficiently clear and sufficiently significant advance to lifelong RL.

As an additional note, the setting under investigation here is not the full lifelong learning setting.  E.g., several of the challenges outlined by Schaul et al. [1] are not treated, and this work is, instead, situated in a somewhat typical multi-task setting with substantlal structure.  That is not bad, but it would be good if this is reflected clearly in all the statements, and, e.g., in the title of the work.

The authors are encouraged to carefully take the provided feedback and see how they can use it to improve their work.  This is an important research direction.  It was just felt the current submission was not quite ready for publication yet.

[1] https://arxiv.org/abs/1811.07004